# Understanding Generative Recommendation with Semantic IDs from a Model-scaling View

## Abstract

Recent advancements in generative models have allowed the emergence of a promising paradigm for recommender systems (RS), known as Generative Recommendation (GR), which tries to unify rich item semantics and collaborative filtering signals. One popular modern approach is to use semantic IDs (SIDs), which are discrete codes quantized from the embeddings of modality encoders (e.g. large language or vision models), to represent items in an autoregressive user interaction sequence modeling setup (henceforth, *SID-based GR*). While generative models in other domains exhibit well-established scaling laws, our work reveals that SID-based GR shows significant bottlenecks while scaling up the model; in particular, the performance of SID-based GR quickly saturates as we enlarge each component – the modality encoder, the quantization tokenizer, and the RS itself. In this work, we identify *the limited capacity of SIDs to encode item semantic information as one of the fundamental bottlenecks*. Motivated by this observation, as an initial effort to obtain GR models with better scaling behaviors, we revisit another GR paradigm that directly uses LLMs as recommenders (henceforth, *LLM-as-RS*). Our experiments show that LLM-as-RS paradigm has superior model scaling properties and achieves up to **20%** improvement over the best achievable performance of SID-based GR through scaling. We also challenge the prevailing belief that LLMs struggle to capture collaborative filtering information, showing that LLMs' ability to model user–item interactions improves as LLMs scale up. Our analyses on both SID-based GR and LLMs across model sizes from 44M to 14B parameters underscore the intrinsic scaling limits of SID-based GR and positions LLM-as-RS as a promising path toward foundation models for GR.

## 1 Introduction

Recommender systems (RS) — such as those for products (Wang et al., 2021; Schafer et al., 1999), videos (Gomez-Uribe & Hunt, 2015; Van den Oord et al., 2013), and friends (Sankar et al., 2021; Ju et al., 2025b) — play a pivotal role in tailoring personalized experiences for millions of users and driving enhanced engagement with online platforms. Conventially, RS follows a two-stage pipeline: *retrieval-and-ranking* (Huang et al., 2020; Weller et al., 2025). During the retrieval stage, a large candidate item corpus is first trimmed down to a subset of items, leveraging computationally efficient heuristics or lightweight neural models, which are quickly processed to filter for broad relevance. This is followed by the ranking stage, where fine-grained user preferences are produced, utilizing complex features that capture intricate patterns, enabling the system to precisely score and order the candidate items (Wang et al., 2021; Zhou et al., 2018).

In parallel, generative recommendation (GR) is emerging as a transformative paradigm of RS, attracting significant attention (Wang et al., 2023; Rajput et al., 2023; Wang et al., 2024) and seeing wide adoption in real-world applications (Zhai et al., 2024; Deng et al., 2025; Tandon, 2025). Unlike conventional methods, GR capitalizes on recent advancements in generative models (Guo et al., 2025; Yang et al., 2025a) and directly generates the recommendation with *just one stage*. This new vertical either directly generates text descriptions of recommended items (Geng et al., 2022b; Tan et al., 2024; Hua et al., 2023; Bao et al., 2023) or uses pre-trained models to extract rich semantic representations that encode broad, open-world knowledge (Yuan et al., 2023; Ren et al., 2024; Yang et al., 2024b). This single-stage approach not only provides additional knowledge from pre-trained models but also avoids the complexity and potential errors associated with multi-stage systems.

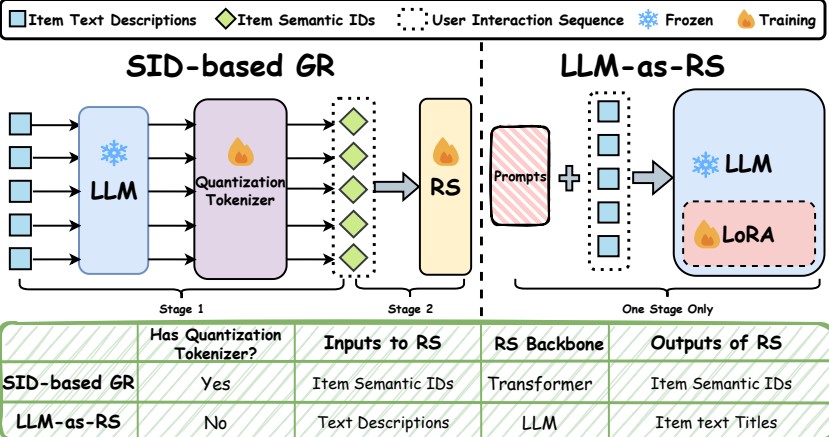

Figure 1: Two GR paradigms we investigate in this paper. SID-based GR first transforms the item textual descriptions into semantic IDs and then trains a transformer to predict the SIDs of the next item, while LLM-as-RS directly takes in the texts and outputs the title of the next item.

Specifically, in GR, semantic IDs (SIDs) (Rajput et al., 2023; Yang et al., 2024b; Deng et al., 2025; Luo et al., 2024; Zheng et al., 2025) are a widely adopted mechanism for integrating existing powerful foundation models with RS. As Figure 1 illustrates, this paradigm consists of two main stages. First, a modality encoder (e.g., LLM or VLM) and a quantization tokenizer (e.g., RQ-VAE (Lee et al., 2022), VQ-VAE (Esser et al., 2021), or Residual K-Means (Deng et al., 2025)) convert item content features (such as image or text) into SIDs. Subsequently, a sequential recommender is trained to autoregressively predict the SIDs corresponding to future user interactions based on the history of previously consumed SIDs (Rajput et al., 2023). GR using SIDs aims to unify semantic knowledge from pretrained foundation models with collaborative filtering (CF) signals from user interactions. The overlap between item SIDs inherently captures item content information to leverage priors of semantic similarity, while next-item prediction supervision enables the model to learn user-behavior patterns, which covers the two pivotal types of knowledge for an effective sequential recommender.

Notably, generative models in other domains have been demonstrated to exhibit scaling behaviors, where the performance improves predictably as a function of increased model size, dataset size, and/or computational budget (Kaplan et al., 2020; Aghajanyan et al., 2023; Tian et al., 2024; Cai et al., 2025). Such observation of scaling laws is not only an academic curiosity, but is critical in informing how to efficiently scale next-generation models (Chung et al., 2024; OpenAI, 2023; Yang et al., 2025a), providing a framework for assessing different model architectures and enabling researchers to identify model classes with more favorable properties.

In this work, we investigate the model scaling behaviors of the SID-based GR paradigm. To do so, we experiment with the impact of model scale in both SID-generation (encoder, quantizer) and autoregressive decoding (sequential recommender) using LLMs of different sizes. Importantly, we empirically discover that the recommendation performance saturates very fast as we scale up the GR pipeline from multiple axes. Such counterintuitive finding stands in stark contrast to the well-established scaling laws observed in other generative domains, suggesting that the benefits of scale do not transfer straightforwardly to SID-based GR. This phenomenon suggests that the SID-based paradigm may contain fundamental bottlenecks, preventing the straightforward application of scaling laws and hence underscoring the need for a more nuanced understanding of how scaling law interacts with GR. Our work aims to understand the following fundamental research questions:

***What are key bottlenecks preventing SID-based GR models from scaling up? Are there other GR paradigms that can overcome them and exhibit better scaling behaviors?***

Existing works in GR strengthen the performance by either incorporating external knowledge such as additional CF signals (Kim et al., 2024; Liao et al., 2024) or richer item semantic information from LLMs (Yang et al., 2024a; 2025b). Beyond these approaches, little attention has been given to whether GR models can be improved simply by scaling up their parameters to better capture item and user information. Drawing on the experience of LLMs, we argue that scaling core architectural components (Kaplan et al., 2020; Brown et al., 2020) represents a promising direction toward more performant GR models and, ultimately, recommendation foundation models. We delve into this direction by conducting a comprehensive analysis on the effects of scaling up GR models and summarize our contributions as follows:

- We show that scaling sequential recommenders in SID-based GR models results in early performance saturation, while neither the LLM encoder nor the quantization tokenizer exhibits scaling behaviors. Our analysis further reveals that the fundamental bottleneck of SID-based GR models lies in their limited capacity to capture item content information. In particular, the SID itself constitutes the key constraint, preventing the knowledge embedded in powerful LLMs from being effectively transferred into the recommender.

- Built atop our findings above, we revisit the GR paradigm of directly using LLMs as recommenders (dubbed LLM-as-RS as in Figure 1) and challenge the prevailing view that they lack the ability to capture CF signals. Instead, we demonstrate that LLMs are capable of modeling CF information, with this capability improving as model size increases. Consequently, the benefits of incorporating external CF embeddings decrease as the backbone scales up.

- Overall, we conclude that the LLM-as-RS paradigm exhibits superior scaling properties compared to SID-based GR models. Its performance improves consistently with model size, showing no signs of saturation. Furthermore, when being scaled up, LLM-as-RS quickly surpasses SID-based GR and achieves up to a 20% improvement using the same training data.

**Clarification**: We note that *this study does not aim to achieve state-of-the-art performance, nor to claim that one paradigm is universally superior to the other under all circumstances*. Rather, our goal is to provide a thorough understanding of their respective advantages and limitations through the model-scaling behaviors analysis, thereby paving the way toward building effective GR models.

## 2 SCALING LAW FOR GENERATIVE RECOMMENDATION

Given an effective GR model, it should well capture two essential types of information: ***collaborative filtering signals (CF)*** and ***semantic information (SI)*** (Zhou et al., 2020; Wu et al., 2023; Sheng et al., 2024). The former refers to how well the model learns the interaction histories, a cornerstone of conventional RS (Kang & McAuley, 2018; Sun et al., 2019); and the latter refers to how well the model learns the semantic content (e.g., text or image) of items, which recently has shown to be highly effective for recommendation (Rajput et al., 2023; Wu et al., 2023; Geng et al., 2022a). The formal definitions of the two types of the information can be found in Section B.

Hence, we posit that a good (in terms of scaling behaviors) GR model *should exhibit improved ability to capture both CF as well as SI as the model progressively scales up*. To quantitatively examine the capability of GR models in modeling CF and SI, we begin by formulating the scaling law equation under the classical risk decomposition principle (Hoffmann et al., 2022). Specifically, we regard CF and SI as two distinct modalities (Liao et al., 2024) and propose the scaling equation for GR models (Equation (1)), following previous works of scaling laws of multimodal models (Aghajanyan et al., 2023; Tao et al., 2024). Since we focus on model scaling, Equation (1) assumes a fixed amount of training data and characterizes how the overall error ($\mathcal{L}$) varies with the number of model parameters allocated to SI learning ($N_{\text{SI}}$) and CF learning ($N_{\text{CF}}$), where the exact forms of $N_{\text{SI}}$ and $N_{\text{CF}}$ depend on the specific model (they can share same terms since some modules learn both CF and SI). The overall loss ($\mathcal{L}$) consists of three additive terms: the minimal achievable error ($E$), the semantic information error ($\frac{B}{N_{\text{SI}}^b}$), and the collaborative filtering error ($\frac{A}{N_{\text{CF}}^a}$), where $E$, $A$, $B$, $a$, and $b$ are positive and empirically determined parameters, formulated as:

Parameters for Semantic Information

Collaborative Filtering Error

$$\mathcal{L}\left( N_{\text{SI}}, N_{\text{CF}} \right) = E + \frac{A}{N_{\text{SI}}^a} + \frac{B}{N_{\text{CF}}^b}. \tag{1}$$

Parameters for Collaborative Filtering

Semantic Information Error

Minimal Achievable Error

Given the retrieval nature of GR and RS, we follow Cai et al. (2025) and adopt the Miss Rate ($\text{MR}@k$) as the metric for the overall error $\mathcal{L}$, where $\mathcal{L} = \text{MR}@k = 1 - \text{Recall}@k$ and $k$ is an integer. To align with the previous works, we report $\text{Recall}@k$ to represent model performance, which allows Equation (1) to be rewritten as:

$$\text{Recall}@k = R_0 - \frac{A}{N_{\text{SI}}^a} - \frac{B}{N_{\text{CF}}^b}, \tag{2}$$

where $R_0$ denotes the maximal achievable performance ($0 < R_0 < 1$) and is also learned empirically. All subsequent empirical analyses are grounded in Equation (2).

**Datasets.** We conduct our experiments on the Amazon Review datasets (He & McAuley, 2016; Hou et al., 2024). We use three subdatasets: Beauty, Sports and Outdoors, and Toys and Games. Since we focus on model scaling in this research, we fix the training and evaluation data amount for the experiments in this paper. We follow the same train/valid/test dataset splits and pre-processing as Ju et al. (2025a) by default. The dataset and implementation details can be found in Section E.

## 3  INVESTIGATING THE MODEL SCALING BEHAVIORS OF SID-BASED GR

**Setup.** To investigate the scaling properties of the SID-based GR, we use TIGER (Rajput et al., 2023) as a representative architecture. This encoder-decoder model, which autoregressively generates SIDs, has proven to deliver promising performance and hence forms the backbone of many contemporary SID-based GR (Yang et al., 2024a; 2025b; Deng et al., 2025). The training process involves two main stages: (1) an LLM encodes item descriptions into semantic embeddings, which are then quantized into discrete SIDs, and (2) a sequential recommender is trained on user SID sequences to predict the next item's SID. During inference, we generate next-item SIDs using beam search (Algorithm 1). Since the official TIGER implementation is not public, our experiments leverage the open-source framework from GRID (Ju et al., 2025a).

As illustrated in Figure 1, SID-based GR contains three overaching components: the pre-trained frozen LLM encoder ($N_{\text{LLM}}$), the trainable quantization tokenizer ($N_{\text{QT}}$), and the trainable downstream recommender ($N_{\text{RS}}$). Since the LLM encoder and tokenizer process only semantic information, we set $N_{\text{CF}} = N_{\text{RS}}$ and $N_{\text{SI}} = N_{\text{RS}} + \gamma_1 N_{\text{LLM}} + \gamma_2 N_{\text{QT}}$, where $0 \leq \gamma_1, \gamma_2 \leq 1$ are effective parameter coefficients of the LLM encoder and quantization tokenizer (since they are frozen during the training stage of RS), respectively. Hence, Equation (2) can be reformulated as:

$$\text{Recall@}k = R_0 - \underbrace{\frac{A}{(N_{\text{RS}} + \gamma_1 N_{\text{LLM}} + \gamma_2 N_{\text{QT}})^a}}_{\text{Semantic Information Error}} - \underbrace{\frac{B}{N_{\text{RS}}^b}}_{\text{Collaborative Filtering Error}}. \quad (3)$$

In the following, we individually examine how model performance varies with $N_{\text{RS}}$, $N_{\text{LLM}}$, and $N_{\text{QT}}$. In Section 3.1, we first show that the sequential recommender quickly reaches a performance plateau as we progressively scale up $N_{\text{RS}}$, indicating that its scaling capabilities are limited. Then in Section 3.2 and Section 3.3, we demonstrate that neither the LLM encoder $N_{\text{LLM}}$ nor the quantization tokenizer $N_{\text{QT}}$ exhibits meaningful scaling behavior on its own within this specific framework. Combining these takeaways, in Section 3.4, our studies reveal that the primary bottleneck to scaling up SID-based GR is its limited ability of encoding semantic information, which is a key component of the model's success. These observations highlight a fundamental limitation of the model scaling behaviors of SID-based paradigm.

### 3.1  SCALING UP THE RS MODULE ($N_{\text{RS}}$) QUICKLY SATURATES THE PERFORMANCE

To start with, we examine how performance scales with the sequential recommender (i.e., the encoder-decoder transformer of TIGER). Specifically, we enlarge the recommender model size ($N_{\text{RS}}$ in Equation (3)) and plot how the performance changes for each dataset. In the experiments, we vary $N_{\text{RS}}$ from 336K to 192M. We utilize Flan-T5-Large (Chung et al., 2024) as the LLM encoder (i.e., $N_{\text{LLM}} = 783\text{M}$) and explore RQ-VAE (Lee et al., 2022) as QT to generates SIDs with codebooks of shape $3 \times 256$ (detailed in Section F.2), following the community convention (Rajput et al., 2023; Ju et al., 2025a). More details of the experiments can be found in Section F.

**Results.** The scaling behvaviors are shown in Figure 2 (please refer to Section H for the results of fitting scaling equation). On all the three datasets, the model performance keeps increasing when $N_{\text{RS}}$ is small. However, when $N_{\text{RS}}$ is larger than 13M, keeping enlarging it does not bring any gains. This means that the model scaling already reaches its upper limit. Hence, we make our first observation:

*Observation 1*: **The RS module exhibits scaling behaviors at small scales (i.e., $N_{\text{RS}} \leq 10^7$), but the performance saturates fast under fixed-data regime.**

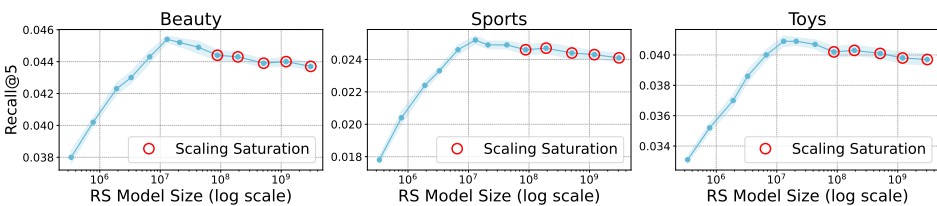

Figure 2: The recommendation performance with varying RS model sizes ($N_{\text{RS}}$). The performance quickly saturates as $N_{\text{RS}}$ scales up to $10^7$ parameters. Shadow: error bar.

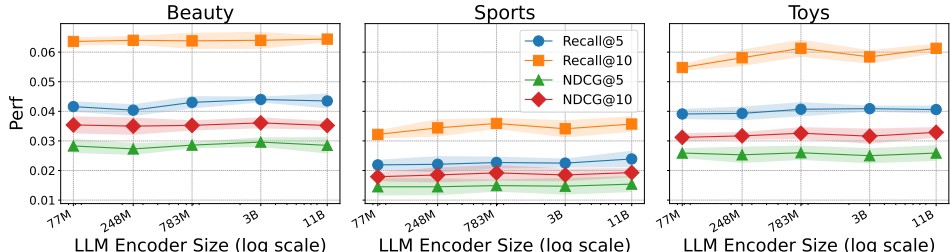

Figure 3: The recommendation performance with varying LLM encoder sizes ($N_{\text{LLM}}$). Little to no effective scaling behaviors are observed. Shadow: error bar.

## 3.2 SCALING UP THE LLM ENCODER ($N_{\text{LLM}}$) BRINGS LITTLE PERFORAMNCE BENEFITS

Based on observations in Section 3.1, a key question is *whether we can overcome the scaling saturation of $N_{RS}$ by increasing $N_{LLM}$ or $N_{QT}$?* In other words, do larger encoders or tokenizers produce more informative item SIDs, which in turn improve the scaling properties of RS? Our scaling law formulations imply this is true if the coefficients $\gamma_1, \gamma_2 > 0$ in Equation (3).

We first examine the scaling behavior of the LLM encoder, which transforms item text descriptions into semantic embeddings. To scale up the encoder, we employ the Flan-T5 series (Chung et al., 2024) with varying sizes ranging from 77M to 11B parameters. Following the optimal setup in the previous section, we fix the number of codebooks in QT at 3, the size of each code book at 256, and $N_{\text{RS}}$ at ~13M. The isolation of variables allows us to pinpoint the specific contribution of $N_{\text{LLM}}$ or $N_{\text{QT}}$. We hypothesize that a larger, more capable LLM should generate richer, more semantically meaningful SIDs, which in turn could enhance the recommendation system's performance.

**Results.** Our analysis, based on metrics like Recall@5, Recall@10, NDCG@5, and NDCG@10, reveals a surprising and critical finding: scaling the LLM encoder size provides negligible to no performance improvement across all three datasets. As seen in our scaling curves, the performance plateaued almost immediately, with no significant gains from using an 11B parameter model over a much smaller 77M parameter model. This suggests that the semantic embedding quality generated by the LLM doesn't act as a bottleneck for the system's performance. The item SIDs produced by even the smallest LLM seem to be sufficiently rich for the task, and adding more parameters doesn't significantly enhance their informativeness. Additional results for other LLMs are reported in Section G with similar observations. We therefore summarize the following observation:

*Observation 2*: **The LLM encoder (i.e., $N_{\text{LLM}}$) in the SID-based GR paradigm shows little to no scaling behaviors (i.e., $\gamma_1 \approx 0$ in Equation (3)).**

## 3.3 SCALING UP THE QUANTIZED TOKENIZER ($N_{\text{QT}}$) QUICKLY SATURATES THE PERFORMANCE AS WELL

Having shown that scaling up LLM encoders ($N_{\text{LLM}}$) does not overcome performance saturation, we next investigate if enlarging the quantized tokenizer ($N_{\text{QT}}$) is more effective. The RQ-VAE tokenizer architecture (Zeghidour et al., 2021; Lee et al., 2022) provides two primary scaling dimensions (detailed in Section F.2): (1) increasing the number of codebooks and (2) increasing the codebook size (cardinality). Our experiments evaluate both approaches. In the following experiments, we evaluate both approaches. Specifically, we fix the codebook size at 256 when varying the number of codebooks, and fix the number of codebooks at 3 when varying the size of each codebook.

Note that modifying the tokenizer's configuration also changes the total SID vocabulary, which directly impacts the embedding table size of the subsequent RS transformer. To account for this

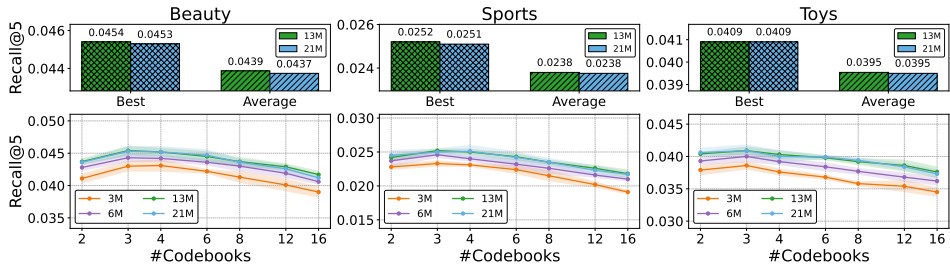

Figure 4: *Lower*: Scaling behaviors of quantization tokenizer when varying the number of codebooks. *Upper*: Comparison of performances between RS modules of 13M and 21M parameters. Overall, increasing the number of codebooks does not overcome the scaling saturation. Shadow:error bar.

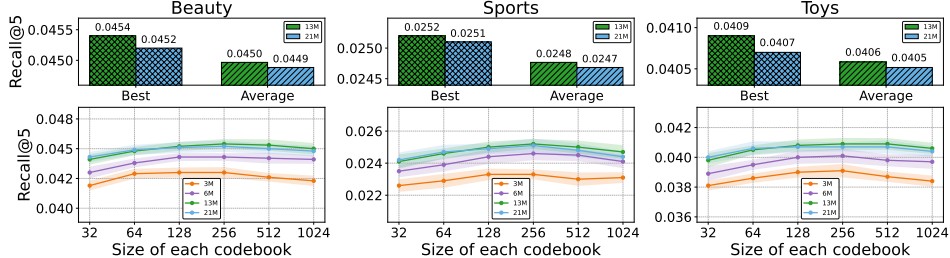

Figure 5: *Lower*: Scaling behaviors of the quantization tokenizer when varying the size of each codebook. *Upper*: Comparison of performances between RS modules of 13M and 21M parameters. Increasing the the size of each codebook does not overcome the scaling saturation. Shadow: error bar.

and avoid suboptimal scaling behaviors (Tao et al., 2024), we scale the RS module in tandem with the tokenizer. Specifically, we vary the number of codebooks from 2 to 16 (while fixing codebook size at 256) and the codebook size from 32 to 1024 (while fixing the number of codebooks at 3). Concurrently, we scale the non-embedding parameters of the RS module from 3M to 21M. To ensure the input features are sufficiently expressive, we use the largest LLM encoder (i.e., 11B).

**Results.** We investigate how performance is affected by scaling the quantization tokenizer, varying both the number of codebooks in Figure 4 and their sizes in Figure 5. This analysis is driven by two central questions: (1) *Can performance gains be achieved by scaling the tokenizer in isolation?* (2) *Does jointly scaling the tokenizer with the RS yield further improvements?*

Accordingly, we plot performance against both the number and size of codebooks, while also comparing RS modules of different sizes (13M and 21M parameters). The results show that the framework achieves optimal performance when the number of codebooks is 3 and the size of each codebook is 256. Beyond these settings, enlarging either dimension generally leads to performance degradation. This indicates that any marginal gains from larger tokenizers are outweighed by the increased learning difficulty for the sequential recommender associated with longer and more complex SIDs. Moreover, the performance comparison reveals no significant advantage of the 21M RS module over the 13M module, suggesting that scaling the tokenizer does not effectively overcome the saturation of the sequential recommender. In summary, we draw the following observation:

***Observation 3*: The quantization tokenizer exhibits scaling behaviors at small scales (i.e., $3\times256$) but further scaling will result in little gains or even performance drops ($\gamma_2 \approx 0$ in Equation (3)).**

### 3.4 ARE SIDS THE BOTTLENECK? AN ABLATION STUDY

From the experiments above, we observe that the scaling of the SID-based GR model exhibits a fundamental bottleneck – neither scaling the LLM encoder nor enlarging the quantization tokenizer improves the quality of the SIDs, and little to no scaling behavior emerges. ***This suggests the bottleneck may not lie in the model's individual components but in the architectural design of the SID-based GR itself.*** In this subsection, we ask: *what will happen if we bypass SIDs and directly input semantic information into the RS module?* We hypothesize that the SID itself could constitute a key bottleneck for scaling the entire framework, as distilling dense LLM embeddings into discrete SIDs discards substantial semantic information, preventing the RS from fully exploiting the LLM's knowledge.

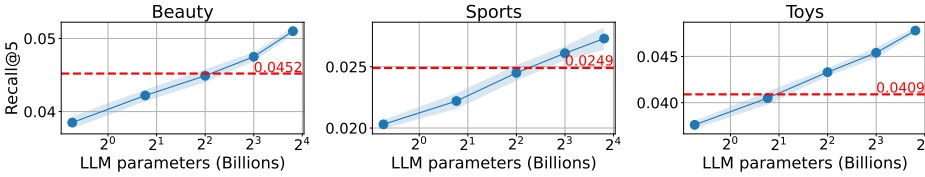

Figure 7: The general scaling behaviors of the LLM-as-RS model ($N_{\text{LoRA}} \approx 0.01 N_{\text{LLM}}$). The Red dashed lines are the best scaling performance of the SID-based model. Shadow: error bar.

To validate this hypothesis, we conduct an ablation study by introducing external knowledge sources of collaborative filtering and semantic information. Specifically, we inject collaborative filtering (CF) embeddings and LLM embeddings into the RS module respectively, and observe the resulting performance changes. For CF embeddings, we train a SASRec model (Kang & McAuley, 2018) on the same training set for each dataset and use its learned item embeddings as CF embeddings. The CF embeddings can be viewed as a knowledge source of user behaviors since the SASRec only accesses the user interaction sequences. For LLM embeddings, we directly employ the original Flan-T5-xxl encoder outputs as item embeddings. The LLM embeddings can be viewed as a knowledge source of item contents since the LLM encoder only accesses the item content descriptions.

We integrate these external embeddings into the RS module as follows: for the SID of item $i$, after passing through the RS embedding table, we obtain an item embedding $e_i$. The corresponding CF or LLM embedding is then processed through a trainable MLP adaptor and added to $e_i$. The subsequent training procedure is identical to that of the standard RS module in TIGER. For more details of the experiments, please refer to Section F.

**Results.** The results are presented in Figure 6, where we evaluate RS modules with 13M and 21M parameters. We find that injecting additional CF embeddings has little effect on performance, suggesting that the framework already captures collaborative filtering signals well and the injected embeddings provide largely redundant information. In contrast, injecting LLM embeddings directly into the RS module yields substantial performance improvements. Moreover, the gains are more pronounced for the 21M RS module than for the 13M variant, indicating that the saturation of scaling is alleviated when richer semantic information is available. Overall, these results

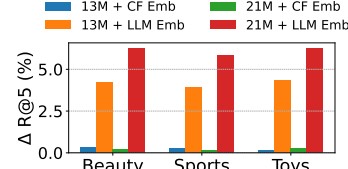

Figure 6: Recall@5 gains from CF/LLM embeddings. Injecting LLM embeddings to RS module break the scaling bottleneck.

demonstrate that the scaling bottleneck arises from the limited semantic information encoded in SIDs. Scaling the encoder or tokenizer fails to produce more informative SIDs, underscoring a fundamental limitation of scaling SID-based GR models. We therefore summarize the following observation:

*Observation 4*: **The poor scaling of the SID-based GR lies in architectural design of the SID-based GR, constrained by its limited ability to extract semantic information using SIDs.**

## 4 BEYOND THE LIMITATIONS OF SIDs: SCALING UP LLM-AS-RS MODEL

We have demonstrated in the previous section that SID-based GR suffers from intrinsic scaling bottlenecks, primarily due to its sub-optimal ability to effectively leverage the rich open-world knowledge encoded in LLMs. Hence, in this section, we turn to an alternative paradigm that directly employs LLMs as recommender (LLM-as-RS), to examine whether they can overcome the scaling bottlenecks. We adopt the vanilla form of LLM recommender (Figure 1). The inputs are plain texts, consisting of simple prompts that describe the task and a list of items' text descriptions representing the user interaction sequence. Unlike some previous works (Liao et al., 2024; Kim et al., 2025), *our prompts do not contain candidate sets of the next item to ensure a fair comparison*. During the training stage, the model is fine-tuned through the LoRA algorithm (Hu et al., 2022). during inference, constrained beam search (Algorithm 1) is used to generate the textual titles of next item. More details of the model and prompt can be found in Section D.

As shown in Figure 1, the parameters of the model consist of two parts: the frozen LLM weights ($N_{\text{LLM}}$) and the trainable LoRA weights ($N_{\text{LoRA}}$). We set $N_{\text{SI}} = N_{\text{LoRA}} + \gamma N_{\text{LLM}}$ and $N_{\text{CF}} = N_{\text{LoRA}} + \beta N_{\text{LLM}}$, where $0 \le \gamma, \beta \le 1$ are the effective parameter coefficients of the frozen LLM to learn semantic information and collaborative filtering signals, respectively. Hence, Equation (2) can be re-written as Equation (4):

Figure 8: Fitting Equation (4) to the empirical data yields high $R^2$ values, indicating strong goodness of fit. $\gamma, \beta > 0$ and that the frozen LLM contributes to the learning of both CF and SI.

$$\text{Recall@}k = R_0 - \underbrace{\frac{A}{(N_{\text{LoRA}} + \gamma N_{\text{LLM}})^a}}_{\text{Semantic Information Error}} - \underbrace{\frac{B}{(N_{\text{LoRA}} + \beta N_{\text{LLM}})^b}}_{\text{Collaborative Filtering Error}}. \tag{4}$$

In the following, we first demonstrate that the LLM-as-RS paradigm exhibits superior scaling properties compared to the SID-based GR (Section 4.1). Furthermore, we show that scaling up the LLM in LLM-as-RS setup enhances its ability to model collaborative filtering signal, challenging the prevailing view of this paradigm (Section 4.1 and Section 4.2).

### 4.1 Investigating the Scaling Behaviors of LLM-as-RS

This subsection investigates the scaling properties of LLM-as-RS by comparing its performance to that of a SID-based GR under the same data budget. Specifically, we simultaneously scale both the trainable and frozen weights and plot the resulting scaling curves. Our experiments use the Qwen3 model series, with sizes ranging from 0.6B to 14B parameters. Throughout these experiments, the trainable LoRA weights are maintained at approximately 1% of the total LLM parameters.

As shown in Figure 7, simple LLM-as-RS models can substantially outperform SID-based GR (highlighted in Red) when scaled up. Moreover, the performance consistently improves as the model scales, with no signs of saturation within the test dataset size range, suggesting that the LLM-as-RS exhibits stronger scaling behaviors than the SID-based GR.

Furthermore, we analyze the respective contributions of the LoRA and LLM weights to scaling. In particular, we aim to address the question: does scaling the frozen LLM weights enhance the model's ability to learn both user behaviors and item contents? To provide a quantitative answer, we fit Equation (4) and estimate the empirical values of $\gamma$ and $\beta$. Specifically, we vary the LoRA rank within $\{8, 16, 24, 32, 40\}$ and the LLM size within $\{0.6B, 1.7B, 4B, 8B, 14B\}$. This yields $5 \times 5 = 25$ data points covering different combinations of LoRA and LLM sizes. Following prior work (Hoffmann et al., 2022), we employ Huber loss (Huber, 1992) with the L-BFGS optimizer (Nocedal, 1980) to estimate the parameters $(R_0, A, B, \gamma, \beta, a, b)$ as:

$$\min_{R_0, A, B, \gamma, \beta, a, b} \sum_{\text{Runs } i} \text{Huber}_{\sigma=0.03} \left[ \log \hat{\mathcal{R}}(N_{\text{LLM}}, N_{\text{LoRA}}) - \mathcal{R}_i \right], \tag{5}$$

where $\hat{\mathcal{R}}$ and $\mathcal{R}$ are the predicted and experimental values for Recall@k, respectively. We show the fitting results in Figure 8, revealing two key findings. First, the high R-square (coefficient of determination) values indicate a strong fit of the equation to the data, confirming the effectivenss of Equation (4). Second, both the LoRA weights and the frozen LLM demonstrate clear scaling behaviors, as evidenced by their positive scaling exponents (i.e., both $\gamma$ and $\beta$ are larger than 0). Further details on the fitting results are available in Section I.

***Observation 5.*** **In the LLM-as-RS paradigm, both the trainable LoRA weights and frozen LLM weights show scaling behaviors. Moreover, the frozen LLM has scaling behaviors on learning both semantic information and collaborative filtering signals ($\gamma, \beta > 0$ in Equation (4)).**

### 4.2 Can LLM-as-RS Exihibit Scaling Behaviors for CF Signals?

Moreover, we further check the model scaling behaviors of LLM-as-RS on learning collaborative filtering signals. To achieve this, we incorporate external CF embeddings, a common technique

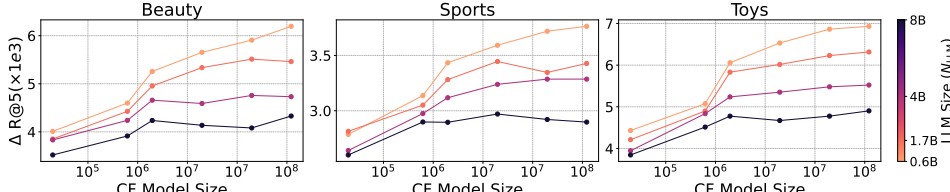

Figure 9: Scaling behaviors of injecting external CF embeddings. The y-axis metric ($\Delta$Recall@5) measures the performance differences between models with or without external CF embeddings. For a fixed CF model size, the performance gains contributed by its CF embeddings decrease as the LLM backbone size increases, indicating that LLM-as-RS exhibits scaling behaviors for CF signals.

for integrating such information into LLM-based recommender systems (Kim et al., 2024; Liao et al., 2024; Kim et al., 2025). Similar to the methodology Section 3.4, we use item embeddings from pre-trained SASRec (Kang & McAuley, 2018) models as the source of these CF signals. These embeddings are first processed by a trainable MLP adapter and then concatenated with the corresponding item's token embeddings within the LLM (see Figure 19). We conduct two sets of experiments to analyze scaling trends: first, by varying the size of the SASRec model that generates the CF embeddings, and second, by varying the LLM size ($N_{\text{LLM}}$). To ensure a fair comparison, we keep $N_{\text{LoRA}}$ fixed at ~50M for all experiments. Experimental details are available in Section F.

**Results.** The scaling curves are shown in Figure 9 with evidence supporting that $\gamma$ in Equation (4) is larger than 0. We further prove this by contradiction. Assuming $\gamma = 0$, then Equation (4) becomes

$$\text{Recall@}k = R_0 - \frac{A}{(N_{\text{LoRA}} + \gamma N_{\text{LLM}})^a} - \frac{B}{N_{\text{LoRA}}^b}. \tag{6}$$

After adding the external CF embeddings and adapter, the scaling equation becomes

$$\text{Improved Recall@}k = R_0 - \frac{A}{(N_{\text{LoRA}} + \gamma N_{\text{LLM}})^a} - \frac{B}{(N_{\text{LoRA}} + N_{\text{SA}})^b}, \tag{7}$$

where $N_{\text{SA}}$ is the total number of parameters of the SASRec model and the adapter. Hence we can get the scaling equation for the performance gains by Equation (7) minus Equation (6), written as

$$\Delta\text{Recall@}k = \frac{B}{N_{\text{LoRA}}^b} - \frac{B}{(N_{\text{LoRA}} + N_{\text{SA}})^b}. \tag{8}$$

Since Equation (8) does not involve $N_{\text{LLM}}$ and we fix $N_{\text{LoRA}}$, the scaling curves with different $N_{\text{LLM}}$ should overlap. However, the actual curves clearly diverge, indicating that the assumption is invalid and $\beta \neq 0$. Furthermore, we observe that larger SASRec models generally yield larger gains, consistent with prior findings (Zhang et al., 2024b). Yet, the performance gains of a fixed-size SASRec model decrease as the LLM backbone scales up. This qualitatively demonstrates that increasing the frozen LLM size already enhances the model's capacity to capture CF signals, implying $\beta > 0$. We also confirm a similar phenomenon when injecting CF embeddings through an alternative method, as shown in Section J.

*Observation 6.* **Gains from external CF embeddings decrease as LLM backbone scales up, proving that $\beta > 0$ in Equation (4) and LLM-as-RS exihibits scaling behaviors for CF signals.**

## 5 CONCLUSIONS

In this work, we examined the scaling behaviors of current mainstream paradigms in generative recommendation. Our work is the first to show that SID-based GR has an essential bottleneck on utilizing semantic information, limiting the scaling of the whole framework with model capacity. In contrast, the LLM-as-RS paradigm can better learn both collaborative filtering signals as well as semantic information, making it a more favorable candidate for recommendation foundation models. Nonetheless, LLM-as-RS currently suffers from efficiency limitations, as discussed in Section K. Specifically, SID-based GR remains preferable when budgets are constrained and efficiency is critical, whereas LLM-as-RS is more suitable when performance is prioritized and resources are sufficient. More discussions on limitations are in Section N. Overall, our findings suggest that how to overcome the shortcomings of current paradigms and fully unlock the potential of LLMs is a critical topic for further research.

## ETHICS STATEMENT

We acknowledge that we have read and commit to adhering to the ICLR Code of Ethics. Our study relies solely on publicly available benchmark datasets such as Amazon Review Dataset (Hou et al., 2024). While our proposed method presents no direct ethical concerns, the improved performance could be leveraged for both ethical and unethical applications involving generative recommendation systems. We emphasize the importance of applying machine learning algorithms responsibly to achieve socially beneficial results.

## REPRODUCIBILITY STATEMENT

Our experiments are based on the public datasets and code (Hou et al., 2024; Ju et al., 2025a). To help reproducibility of the results, we provide experiment settings in Section 3 and Section 4, Section D, and Section F.

## USAGE OF LARGE LANGUAGE MODELS

In this manuscript, we solely utilize LLMs to polish the writing and check grammatical errors. We have reviewed the generated contents provided by large language models and will be responsible for the correctness of the polished content.

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

APPENDIX CONTENTS

# A    RELATED WORKS

Generative recommendation (GR) has recently emerged as a promising direction for unifying content modeling and collaborative filtering through generative paradigms. We study two mainstream GR paradigms: *SID-based GR*, which leverages discrete semantic identifiers (SIDs) derived from modality encoders and tokenizers, and *LLM-as-RS*, which directly employs large language models as recommenders.

**SID-based GR.**    SID-based GR transforms items into short sequences of discrete SIDs and trains sequence models to generate the next item's SID. TIGER Rajput et al. (2023) formulates recommendation as *generative retrieval*: it first derives a *Semantic ID (SID)* for each item from content features via residual quantization and then autoregressively predicts the SID of the next item with an encoder-decoder transformer. COBRA (Yang et al., 2025b) proposes a unified generative recommendation framework that cascades sparse semantic IDs and dense vectors with end-to-end training and BeamFusion inference, achieving both precision and diversity in large-scale recommendation systems As a follow-up to TIGER, LIGER (Yang et al., 2024a) introduces a hybrid retriever that augments SID-based generation with dense text representations, narrowing the gap with dense retrieval while explicitly addressing TIGER's difficulty in generating *cold-start* items. Beyond fixed tokenizers, ETEGRec (Liu et al., 2025) trains the item tokenization and the generative recommender end-to-end, aligning the learned item tokens with the generative objective to reduce tokenization–model mismatch and improve overall accuracy. OneRec (Deng et al., 2025) unifies retrieval and ranking within a single generative framework and applies preference-alignment techniques (e.g., DPO/IPO-style objectives) to better align the generator with ranking metrics, demonstrating that retrieve–rank pipelines can be collapsed into one generative model without sacrificing performance.

**LLM-as-RS.**    A parallel line of work treats a large language model itself as the recommender. IDGenRec (Deng et al., 2025) learns concise, semantically rich *textual IDs* (composed of human-language tokens) and jointly trains a textual ID generator with an LLM-based recommender, enabling seamless text-to-text recommendation. GPT4Rec (Li et al., 2023) reframes recommendation as "*generate search queries, then retrieve*": a fine-tuned GPT-2 produces multiple queries that capture diverse user interests. GenRec (Ji et al., 2024) fine-tunes LLaMA directly to generate the next item name conditioned on a prompted interaction history, showing that a pure text-to-text LLM can act as a generative recommender on standard datasets. To address latency and scalability, LlamaRec (Yue et al., 2023) proposes a two-stage design: a lightweight ID-based sequential model retrieves candidates, and a parameter-efficient LLM ranks them via a verbalizer that maps outputs to candidate scores. LLaRA (Liao et al., 2024) further bridges conventional sequential recommenders and LLMs via a hybrid prompting scheme that fuses textual item features with behavioral tokens projected from ID-based item embeddings, together with curriculum prompt tuning that warms up on text-only prompts before progressively injecting behavioral knowledge. 360Brew (Firooz et al., 2025) constructs a decoder-only large-scale model with a textual interface that handles 30+ ranking/recommendation tasks across the platform with competitive or better offline performance, highlighting the possibility of a single foundation model for broad RS tasks.

**Relation to prior works.**    There has been some exploration of scaling laws in related areas of recommendation and retrieval. For example, Zhang et al. (2024c) investigate the scaling behavior of purely ID-based recommendation models, while Fang et al. (2024) study the scaling laws of generative retrieval. To the best of our knowledge, however, the scaling laws of *generative recommendation* have not yet been systematically examined. In addition, prior works (Sheng et al., 2024; He et al., 2025) suggest that LLMs are effective at capturing collaborative filtering signals, but they primarily employ LLMs as embedding models rather than as generative recommenders. Moreover, these studies do not provide a quantitative framework for analyzing LLMs' collaborative filtering capabilities as we do in this work.

## B  THE FORMAL DEFINITIONS OF CONCEPTS

**Setup.**  Let $\mathcal{U} = \{u_1, \ldots, u_{|\mathcal{U}|}\}$ be the set of users, $\mathcal{I} = \{i_1, \ldots, i_{|\mathcal{I}|}\}$ be the set of items, and $\mathcal{R} \subseteq \mathcal{U} \times \mathcal{I}$ be the observed user–item interactions. For a given user $u \in \mathcal{U}$, let $H_u = \{i \mid (u, i) \in \mathcal{R}\}$ denote the interaction history of $u$. A generative recommendation model aims to predict the next interacted item $J_u \in \mathcal{I}$ via the distribution $P(J_u \mid H_u, \{X_i\}_{i \in \mathcal{I}})$, where $X_i$ denotes the semantic features of item $i$.

**Collaborative Filtering Signals (CF)**  . The collaborative filtering signals correspond to the statistical dependencies embedded in $H_u$ that explain $J_u$, independent of item semantics. Formally, CF information can be expressed as the conditional mutual information

$$\mathsf{CF} \;=\; I(J_u; H_u \mid \{X_i\}_{i \in \mathcal{I}}),$$

which measures how much predictive information about $J_u$ is uniquely contributed by the user's historical interactions.

**Semantic Information (SI).**  Suppose each item $i \in \mathcal{I}$ is associated with content features $\mathbf{x}_i \in \mathcal{X}$, where $\mathcal{X}$ may represent textual, visual, or multimodal embeddings. Let $X_i$ be the random variable corresponding to $\mathbf{x}_i$. The semantic information is the representational knowledge contained in item semantics that explains the likelihood of interaction. Formally, SI is captured by the conditional mutual information

$$\mathsf{SI} \;=\; I(J_u; \{X_i\}_{i \in \mathcal{I}} \mid H_u),$$

which quantifies the additional information provided by item semantics for predicting $J_u$, beyond what is contained in $H_u$.

**Generative Recommendation Task.**  An effective generative recommender (GR) model integrates both signals:

$$P(J_u \mid H_u, \{X_i\}_{i \in \mathcal{I}}),$$

where $H_u$ captures **collaborative filtering information** and $\{X_i\}$ captures **semantic information**. Thus, CF reflects *relational structure* among users and items, while SI reflects *content-driven similarity* among items.

## C    Constrained Beam Search

---

**Algorithm 1** Constrained Beam Search

---

**Input:** Initial context sequence $Y_0 = (y_1, ..., y_c)$, beam width k, max number of new tokens $T_{max}$, a Trie $\mathcal{T}$ containing all valid token sequences, and a generative model that can gernerate the next token of a given sequence.

**Output:** The highest probability valid sequence $Y_{best}$.

1: Initialize beam set $\mathcal{B} \leftarrow (Y_0, score = 0)$                    ▷ **Score is log probability**
2: **for** t←1 **to** $T_{max}$ **do**
3:        Initialize an empty candidate set $C = \{\}$
4:        **for all** sequence $Y_{t-1}$, score $S_{t-1}$ **in** $\mathcal{B}$ **do**
5:                **if** $Y_{t-1}$ ends with an EOS token **then**
6:                        Add $(Y_{t-1}, S_{t-1})$ to $\mathcal{C}$ and continue
7:                Get next-token logits from model: $L \leftarrow \text{Model}(Y_{t-1})$
8:                Let $Y'_{t-1}$ be the generated part of the sequence (after the initial context)
9:                Get allowed next tokens from Trie: $\mathcal{A} \leftarrow \mathcal{T}.\text{get\_allowed\_next\_tokens}(Y'_{t-1})$
10:                **if** $\mathcal{T}.\text{is\_valid\_sequence}(Y'_{t-1})$ **then**
11:                        $\mathcal{A} \leftarrow \mathcal{A} \cup \{\text{EOS\_token}\}$
12:                **if** $\mathcal{A}$ is empty **then**
13:                        $\mathcal{A} \leftarrow \{\text{EOS\_token}\}$
14:                Create a mask $M$ where $M_i = -\infty$ if token $i \notin \mathcal{A}$ and $M_i = 0$ otherwise.
15:                Apply mask: $L' \leftarrow L + M$
16:                Convert to log probabilities: $P'_{log} \leftarrow \text{LogSoftmax}(L')$
17:                **for all** token $y_t$ in vocabulary **do**                    ▷ **Expansion Step: Add new candidates**
18:                        **if** $P'_{log}[y_t] > -\infty$ **then**
19:                                $Y_t \leftarrow \text{concat}(Y_{t-1}, y_t)$
20:                                $S_t \leftarrow S_{t-1} + P'_{log}[y_t]$
21:                                Add $(Y_t, S_t)$ to $\mathcal{C}$
22:        $\mathcal{B} \leftarrow$ Top-k candidates from $\mathcal{C}$ by score                    ▷ **Pruning Step: Select top-k candidates**
23:        **if** all sequences in $\mathcal{B}$ end with an EOS token **then**
24:                **break**
25: **return** sequences from $\mathcal{B}$ (remove the last EOS tokens).

---

ALGORITHM EXPLANATION

This algorithm generates a sequence of tokens one step at a time, keeping track of the k most probable sequences (beams) at each step.

- **Token**: A "token" is a generic term for a discrete unit being generated. For a Large Language Model (LLM), this is a word or sub-word ID. For the TIGER model, this is a hierarchical ID for a specific level (e.g., category, sub-category).

- **Trie (T)**: A prefix tree data structure that efficiently stores all valid sequences from a predefined dictionary. This dictionary could be a set of allowed sentences (for an LLM) or the complete catalog of valid hierarchical item IDs (for TIGER).

- **Constraint Step (Lines 9-16)**: This is the core of the algorithm. Before the model chooses the next token, it consults the Trie to determine the set of all valid next tokens, given the sequence generated so far. It then creates a mask to set the probability of all invalid tokens to zero (by setting their logits to the negative infinity), forcing the model to only consider valid continuations.

- **Expansion & Pruning (Lines 17-22)**: The algorithm explores all valid continuations from the current set of beams and then prunes this expanded set back down to the top k most probable sequences, which become the beams for the next step. This process repeats until a maximum length is reached or all beams have been completed with an End-of-Sequence (EOS) token.

## D  THE IMPLEMENTATION DETAILS OF THE LLM-AS-RS MODEL

In Figure 10, we present the framework of the LLM-as-RS model used in our experiments. The prompts are deliberately simple, consisting of only two sentences. The raw textual descriptions of items are organized into the user interaction sequence. We limit the maximum user interaction sequence length to be 20. During training, we optimize the LoRA weights (Hu et al., 2022) using cross-entropy loss. The LoRA weights are applied on modules including ['k_proj', 'v_proj', 'q_proj', 'o_proj', 'gate_proj', 'up_proj', 'down_proj']. For inference, we employ constrained beam search (Algorithm 1) to guide the model in generating the next-item prediction. Specifically, we construct a trie from all item titles in the dataset to ensure that the model outputs only valid item titles.

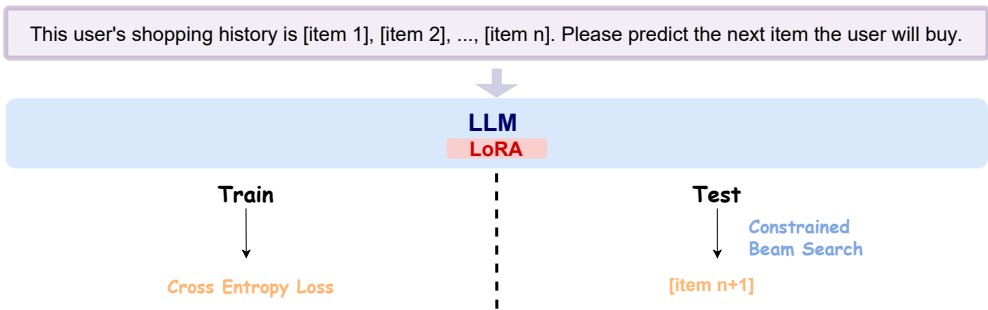

Figure 10: The structure and prompts for the LLM-as-RS model.

## E  DATASET DETAILS

Here we present the details of the datasets we use in the experiments. We main use three datasets from the Amazon Reviews Datasets (Hou et al., 2024). We list the basic statistics of the datasets in table 1.

| Dataset | # users | # items | # actions |
|---|---|---|---|
| *Beauty* | 22,363 | 12,101 | 198,502 |
| *Toys and Games* | 19,412 | 11,924 | 167,597 |
| *Sports and Outdoors* | 35,598 | 18,357 | 296,337 |

Table 1: Statistics of the datasets used in our experiments.

Furthermore, we also show the distributions of the user interaction sequence lengths in Figure 11, 12 and 13, and show the distributions of the user interaction sequence lengths in Figure 14, 15, 16. These statistics will affect the efficiency of our models.

We also deduplicate ite titles in the dataset during data preprocessing stage. Specifically, the duplication rate is minimal when we limit item titles to 10 tokens. And in order to disregard even this minimal collision, we replace the final token of the identical pairs with a different token from their item descriptions respectively. Hence, the reported results are not impacted, and we will be sure to add this context in the paper as we agree it is important to ensure fairness in this comparison.

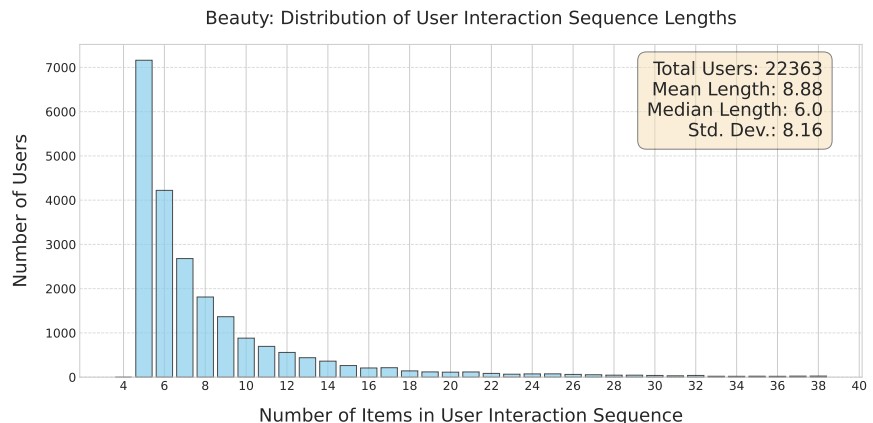

Figure 11: Distribution of user interaction sequence lengths in Beauty dataset.

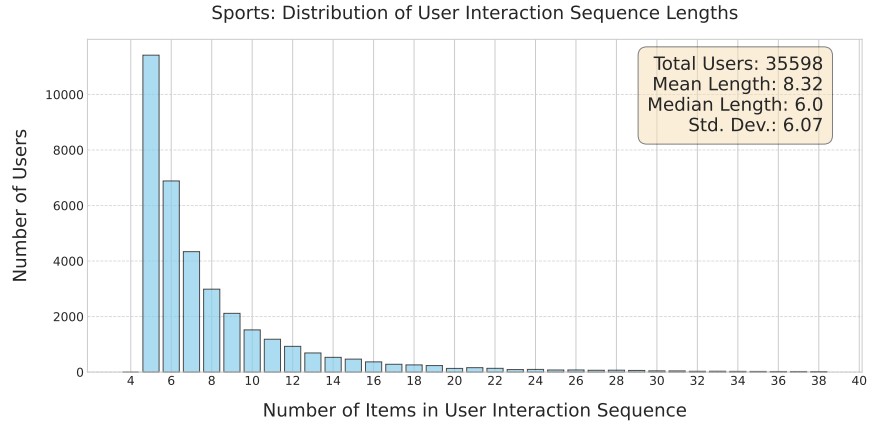

Figure 12: Distribution of user interaction sequence lengths in Sports dataset.

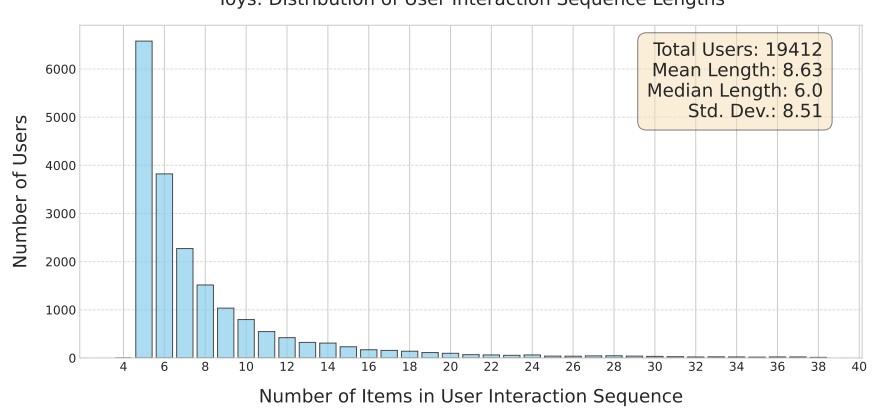

Figure 13: Distribution of user interaction sequence lengths in Toys dataset.

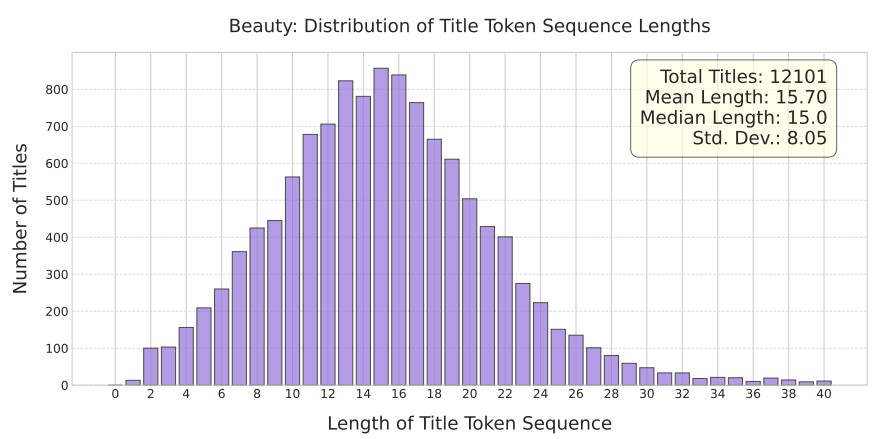

Figure 14: Distribution of title token sequence lengths in Beauty dataset.

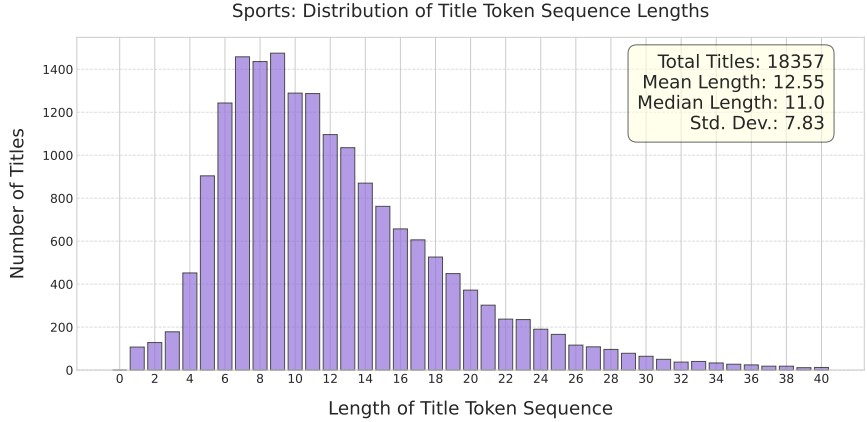

Figure 15: Distribution of title token sequence lengths in Sports dataset.

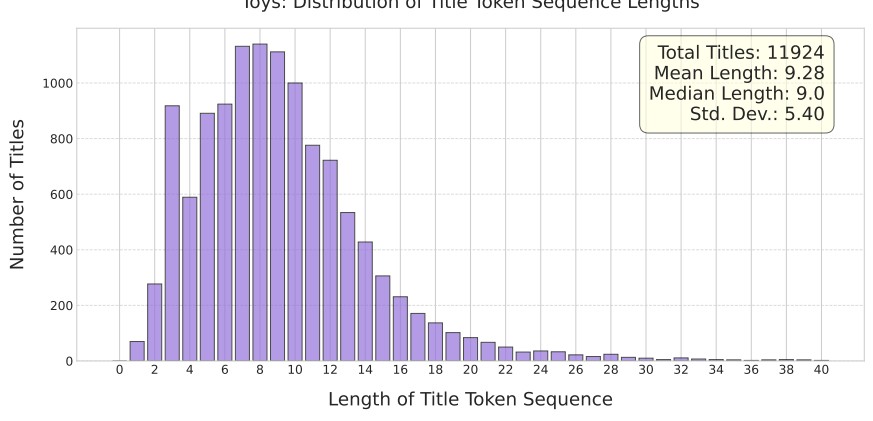

Figure 16: Distribution of title token sequence lengths in Toys dataset.

# F  EXPERIMENT DETAILS

## F.1  GENERAL SETTINGS

We first present the general experimental settings, which are shared across all experiments. Detailed settings specific to each experiment are provided in the corresponding sections. All models are optimized using AdamW (Loshchilov & Hutter, 2017). For each training process, we search the learning rate over {1e-2, 1e-3, 1e-4} and select the best-performing value. All experiments are conducted on 8×80G and 8×40G NVIDIA Tesla A100 GPUs.

## F.2  EXPERIMENT SETTINGS OF SECTION 3

**Quantization tokenizer structure.**  The tokenizer maps a semantic embedding $h_i \in \mathbb{R}^d$ (produced by an encoder $E(\cdot)$) to a sequence of discrete code indices (a *Semantic ID*):

$$\mathrm{SID}_i \;=\; \mathrm{Tokenizer}(h_i) \;=\; \big[\mathrm{SID}_i^{(0)},\, \mathrm{SID}_i^{(1)},\, \ldots,\, \mathrm{SID}_i^{(L-1)}\big],$$

where $L$ is **the number of codebooks (or quantization layers)** and $\mathrm{SID}_i^{(l)}$ denotes the index chosen at layer $l$. In the uniform-size setting each layer has the same **codebook size (cardinality)** $W$, so

$$\mathrm{SID}_i \in \{0, \ldots, W-1\}^L.$$

More generally, allow per-layer sizes $W^{(l)}$ and denote the $l$-th codebook by

$$C^{(l)} = \big\{c_1^{(l)}, \ldots, c_{W^{(l)}}^{(l)}\big\} \subset \mathbb{R}^d, \qquad W^{(l)} \in \mathbb{Z}_{>0}.$$

Then

$$\mathrm{SID}_i \in \prod_{l=0}^{L-1} \{0, \ldots, W^{(l)} - 1\}.$$

For residual (sequential) quantization one typically sets $r_i^{(0)} = h_i$ and for each layer $l = 0, \ldots, L-1$ chooses

$$\mathrm{SID}_i^{(l)} \;=\; \arg\min_{j \in \{1, \ldots, W^{(l)}\}} \big\| r_i^{(l)} - c_j^{(l)} \big\|_2, \qquad r_i^{(l+1)} \;=\; r_i^{(l)} - c_{\mathrm{SID}_i^{(l)}}^{(l)}.$$

The reconstructed embedding is the sum of selected codewords:

$$\hat{h}_i \;=\; \sum_{l=0}^{L-1} c_{\mathrm{SID}_i^{(l)}}^{(l)}.$$

Common compact notation for a uniform codebook configuration is $(L, W)$ (e.g. $(3, 256)$ or $3 \times 256$ meaning $L = 3$ layers each with $W = 256$ entries); for the nonuniform case one writes the vector of sizes $(W^{(0)}, W^{(1)}, \ldots, W^{(L-1)})$. **We use the number of the codebooks and the size of each codebook for $L$ and $W$ in the main text, respectively.**

**Scaling RS modules**  For all the experiments with the SID-based GR model, we keep the length of input user interaction sequence at 20. For the scaling experiments in Section 3.1, we enlarge the RS module according to the Section F.2.

**Injecting CF/LLM embeddings**  For the experiments in Section 3.4, we fetch and inject the LLM embeddings as shown in Figure 17, and produce and inject the CF embeddings as shown in Figure 18.

## F.3  EXPERIMENT SETTINGS OF SECTION 4

For experiments in Section 4.2, we produce and inject the external CF embeddings as shown in Figure 19. And we scale up SASRec models according to Section F.3

| #Params | #Layers | $d_{model}$ | #Heads | $d_{kv}$ | $d_{ff}$ |
|---|---|---|---|---|---|
| 336,000 | 1 | 64 | 3 | 64 | 512 |
| 778,000 | 2 | 64 | 3 | 64 | 512 |
| 1,900,000 | 5 | 64 | 3 | 64 | 512 |
| 3,300,000 | 9 | 64 | 3 | 64 | 512 |
| 6,700,000 | 3 | 128 | 6 | 64 | 1024 |
| 13,000,000 | 4 | 128 | 6 | 64 | 1024 |
| 21,000,000 | 7 | 128 | 6 | 64 | 1024 |
| 43,000,000 | 8 | 192 | 9 | 64 | 1536 |
| 88,000,000 | 9 | 320 | 15 | 64 | 2560 |
| 192,000,000 | 20 | 384 | 18 | 64 | 3072 |

Table 2: The details of scaling RS module in SID-based GR. The #Params are just rounded values. The #Layers are the same for both encoder and decoder in the module.

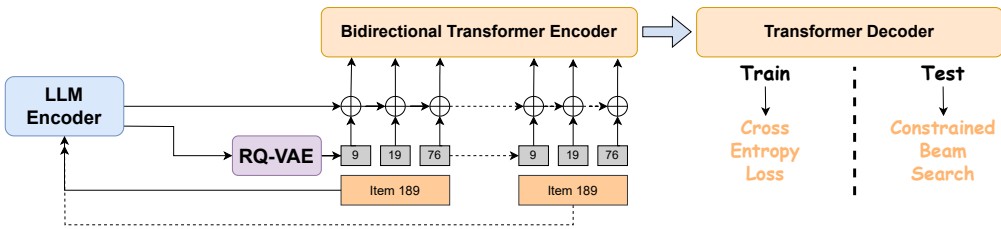

Figure 17: The illustration of injecting the LLM embeddings into SID-based GR model.

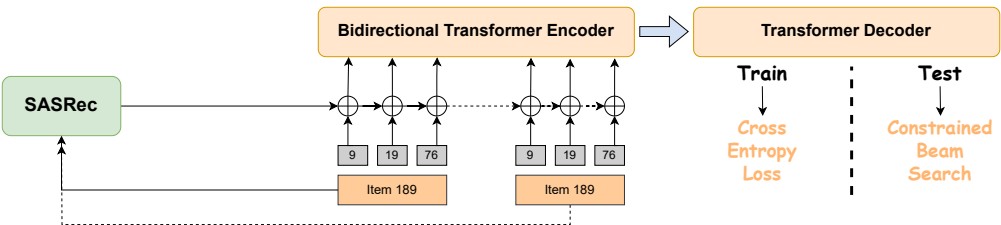

Figure 18: The illustration of injecting the CF embeddings into SID-based GR model.

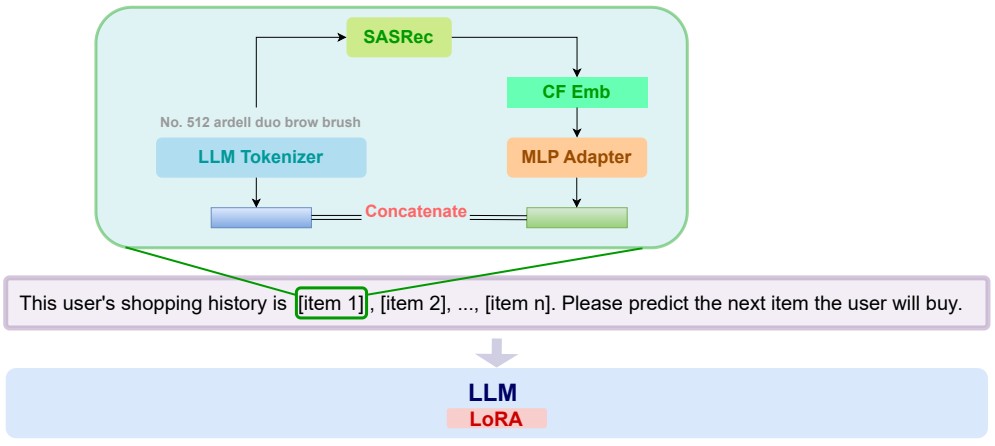

Figure 19: The illustration of injecting the CF embeddings into LLM-as-RS model by Concatenation.

## G SCALING BEHAVIORS OF LLM ENCODERS OF OTHER TYPES

In Section 3.2, we demonstrate that the T5 series models (encoder–decoder architectures) exhibit no scaling behavior when used as the LLM encoder. To strengthen this argument, we further evaluate other types of LLMs as encoders, including encoder-only DeBERTa models (He et al., 2020) and decoder-only Qwen3 models (Yang et al., 2025a). Following the same experimental procedures

| #Params | #Layers | $d_{model}$ | #Heads |
|---|---|---|---|
| 98,304 | 2 | 64 | 2 |
| 786,432 | 4 | 128 | 4 |
| 1,572,864 | 8 | 128 | 4 |
| 6,291,456 | 8 | 256 | 8 |
| 25,165,824 | 8 | 512 | 8 |
| 75,497,472 | 24 | 512 | 8 |

Table 3: The details of scaling the SASRec model.

described in Section 3.2, we plot their scaling curves in Figure 20 and Figure 21. The results reveal two key findings: (1) the performance varies across model types, with T5 achieving the strongest results as an encoder, and (2) regardless of the model type, none exhibit scaling behavior, reinforcing our earlier observation in Section 3.2.

**Results with another quantization tokenizer.** To further verify that the scaling bottleneck of the LLM encoder arises specifically with residual vector quantization, we repeat the experiments using an alternative tokenizer. In particular, we adopt RPG (Hou et al., 2025), which employs Product Quantization as the tokenizer and sentence-T5 (Ni et al., 2021) as the LLM encoder. The results are presented in Figure 22. We observe that scaling up the LLM encoder still yields no performance improvement, confirming the generality of our findings.

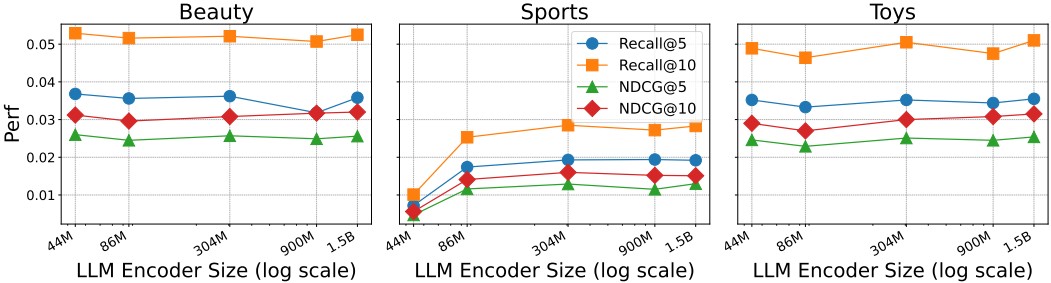

Figure 20: The scaling behaviors of LLM encoder (deberta).

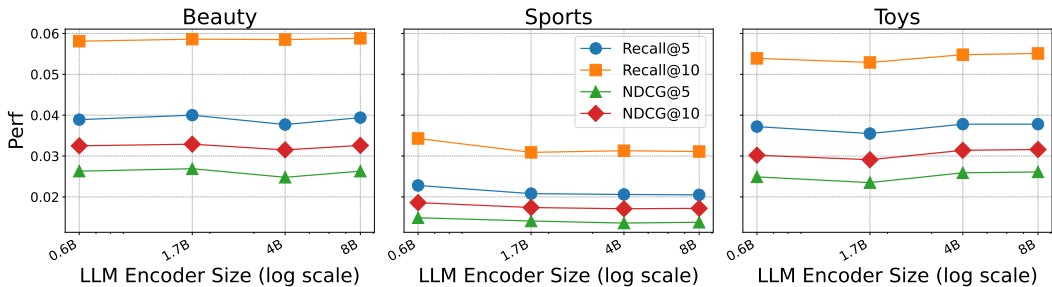

Figure 21: The scaling behaviors of LLM encoder (Qwen3).

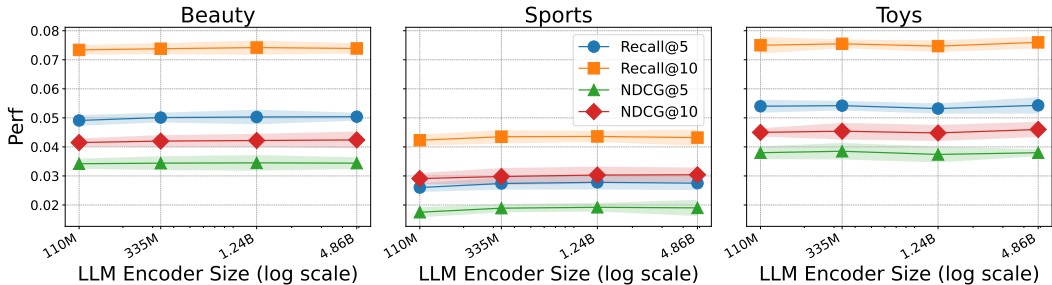

Figure 22: The scaling behaviors of LLM encoder (Sentence-T5) with **Product Quantization**.

## H  FITTING EQUATION 3 TO EMPIRICAL DATA

In the main text, we have shown that $\gamma_1$ and $\gamma_2$ in Equation (3) equal to zero. Hence, Equation (3) can be re-written as

$$\text{Recall@}k = R_0 - \frac{A}{N_{\text{RS}}^a} - \frac{B}{N_{\text{RS}}^b} \tag{9}$$

Then we will fit the equation to the empirical data in the data in Section 3.1, following the same process as Section 4.1 with Equation (10). The results are shown in Table 4. The R-square values are larger than 0.9, indicating the equation fits real data well.

$$\min_{R_0, A, B, a, b} \sum_{\text{Runs } i} \text{Huber}_{\sigma=0.03} \left[ \log \hat{\mathcal{R}}(N_{\text{RS}}) - \mathcal{R}_i \right] \tag{10}$$

|          | Beauty | Sports | Toys   |
|----------|--------|--------|--------|
| R-square | 0.94   | 0.97   | 0.94   |
| $R_0$    | 0.4529 | 3e-1   | 1.7e-1 |
| $A$      | 16.8   | 24.8   | 6.1    |
| $B$      | 1e-2   | 1e-2   | 1e-2   |
| $a$      | 0.6    | 0.63   | 0.52   |
| $b$      | 2.23   | 1.97   | 2.02   |

Table 4: The fitted empirical parameters of scaling laws of SID-based GR.

## I  MORE RESULTS OF THE SCALING LAW FITTING OF THE EQUATION 4

In Section 4.1, we fit Equation 4 with the empirical data. We only report the values of $\gamma$ and $\beta$, here we report the values of all the parameters in Table 5.

Moreover, we compare the held-out errors of Equation 4 and Equation 6 to prove further evidence that $\beta > 0$. Specifically, we randomly choose 20% of the experimental data in Figure 8 as test data, and use the remaining data to fit the two equations. Then we will use the two fitted equations to predict the values of test data and calculate the held-out errors as Zhang et al. (2024a). The errors are listed in Table 6. We find that $\beta > 0$ will lead to less held-out errors, which justify our choice for Equation 4 and further support our argument that $\beta > 0$.

|       | Beauty  | Sports   | Toys    |
|-------|---------|----------|---------|
| $R_0$ | 3e-1    | 3e-1     | 1.7e-1  |
| $A$   | 9.9e2   | 9.87e2   | 2.19e-1 |
| $B$   | 3.4e-1  | 3.35e-1  | 9.93e2  |
| $\gamma$ | 9.08e-2 | 1.82e-2 | 1.74e-1 |
| $\beta$ | 2.10e-2 | 1.69e-2 | 2.29e-2 |
| $a$   | 1.98e1  | 2.02e1   | 2.47e-2 |
| $b$   | 1.39e-2 | 9.47e-3  | 2.02e1  |

Table 5: The fitted empirical parameters in Section 4.1.

|        | $\beta = 0$ | $\beta > 0$ |
|--------|-------------|-------------|
| Beauty | 4.1e-4$\pm$2e-5 | **3.2e-4** $\pm$1e-5 |
| Sports | 1.1e-3 $\pm$3e-5 | **3.4e-4**$\pm$2e-5 |
| Toys   | 1.2e-3$\pm$4e-5 | **9.1e-4**$\pm$2e-5 |

Table 6: Held-out fitting errors (the lower the better) of Equation 4 when $\beta = 0$ and $\beta > 0$.

# J   MORE RESULTS OF INJECTING THE EXTERNAL CF EMBEDDINGS INTO THE LLM-AS-REC MODEL

We further explore an alternative approach to incorporating external CF embeddings into the LLM-as-RS model. The procedure follows Section 4.2, except that, instead of concatenating CF embeddings after the item token embeddings, we directly add them to the item token embeddings, as illustrated in Figure 23. The results in Figure 24 show scaling behaviors consistent with those discussed in the main text, thereby reinforcing our observation.

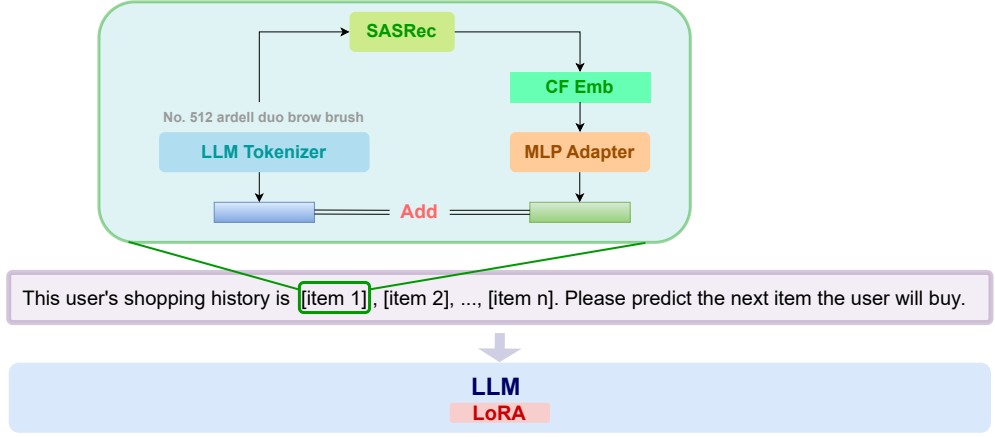

Figure 23: The illustration of injecting the CF embeddings into LLM-as-RS model by ADDING.

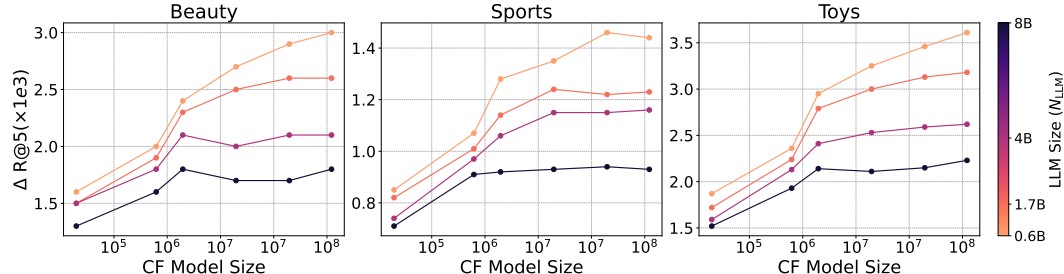

Figure 24: Scaling behaviors of external CF embeddings added to the token embeddings. The y-axis metric ($\Delta$Recall@5) measures the performance differences between models with external CF embeddings and those without. For a fixed CF model size, the performance gains contributed by its CF embeddings decrease as the LLM backbone size increases.

## K  THE COMPUTATION COSTS OF THE TWO PARADIGMS

We compare the training and inference efficiency of the two paradigms on the Beauty dataset, as shown in Figure 25. Training time is measured by the GPU hours required for a model to converge. We observe that under limited training time or computational budgets, the SID-based method performs better. However, as models scale, the performance of LLM-as-RS surpasses that of SID-based GR with comparable training time. On the other hand, inference costs for SID-based GR are substantially lower than those of LLM-as-RS, primarily because SID-based GR generates only four IDs per item, whereas LLM-as-RS must generate full titles, typically longer than four tokens. Thus, SID-based GR maintains efficiency advantages when resources are constrained, while LLM-as-RS becomes preferable when budgets allow and performance is the primary objective.

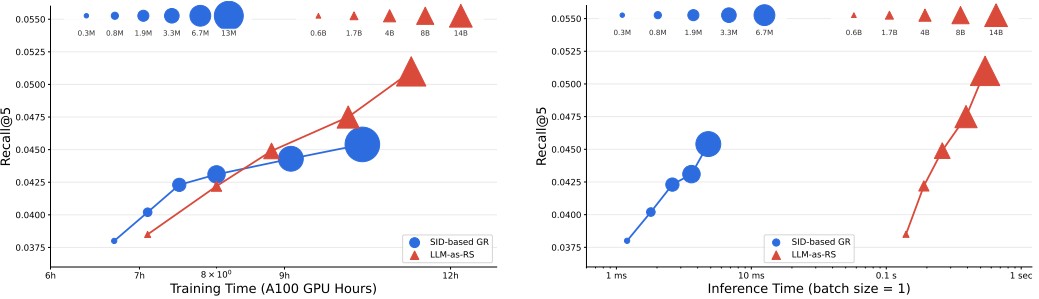

Figure 25: The efficiency comparison between the two paradigms. Left: The training time and performance of models of different sizes. Training time is measured by the GPU hours required for a model to be trained to converge. Right: The inference time and performance of models of different sizes.

## L  INPUTTING SIDS TO LLM-AS-RS MODEL

In this section, we investigate a hybrid of the two paradigms: feeding SIDs generated by the quantization tokenizer into the LLM-as-RS model as user inputs, followed by finetuning the LLM with LoRA. To this end, we repeat the experiments in Figure 7, replacing item titles with item SIDs as inputs to the LLM. We keep the LoRA parameter budget fixed at $N_{\text{LoRA}} = 0.01 N_{\text{LLM}}$, and the results (Recall@5) are reported in Table 7. The results show that although the model exhibits some scaling behaviors, its overall performance remains unsatisfactory. A plausible reason is that the LLM must translate SIDs back into the semantic space, which increases training difficulty. Some works (Chen et al., 2024) are already studying how to better align the SIDs with LLM recommender. We note, however, that this is only a preliminary exploration, and we leave more in-depth investigation for future work.

|        | Qwen3-0.6B | Qwen3-1.7B | Qwen3-4B | Qwen3-8B |
|--------|------------|------------|----------|----------|
| Beauty | 0.0356     | 0.0379     | 0.0393   | 0.0409   |
| Sports | 0.0197     | 0.0217     | 0.0225   | 0.0234   |
| Toys   | 0.0367     | 0.0380     | 0.0392   | 0.0398   |

Table 7: The results when we input SIDs as item representations into the LLM.

## M  COLD-START EXPERIMENTS

In this section, we compare the cold-start ability of the two paradigms. For each dataset, we randomly select 500 items as cold-start items in the test set and remove them from the corresponding training sets. We then train the models and evaluate their Recall@10 on these cold-start items. The results, presented in Table 8, show that LLM-as-RS consistently outperforms SID-based GR, indicating stronger transferability to unseen items.

|        | SID-based GR | LLM-as-RS |
|--------|--------------|-----------|
| Beauty | 0.018        | **0.030** |
| Sports | 0.006        | **0.014** |
| Toys   | 0.022        | **0.028** |

Table 8: The comparison on the cold-start items. LLM-as-RS consistently outperforms SID-based GR.

## N  DISCUSSIONS AND LIMITATIONS

**Modality coverage of item content.** Our experiments represent items solely by *textual* descriptions during encoding and generation. Real-world item contents could be multimodal (e.g., images for products, frames and audio for videos, or speech/music for podcasts). Future work can extend the analysis to image-, audio-, and video-augmented settings (and to cross-modal tokenization schemes) to ascertain whether the scaling ceiling observed with text-only SIDs persists or shifts once additional modalities are introduced.

**Paradigm scope and alternative formulations.** Our scaling observations focus on mainstream instantiations of current paradigms (SID-based GR with residual quantization over text encodings; LLM-as-RS with textual inputs and outputs). Other models might exhibit different scaling behaviors.

**Data scale and dataset composition.** All results were obtained under a fixed-data regime on specific Amazon categories. Switching the dataset domains or scaling the data amount to the industrial level might affect the model scaling behaviors as well.

## O  RESULTS ON THE STEAM DATASETS

To provide more comprehensive results, we repeat the results in Steam dataset (Kang & McAuley, 2018) in the main text and report the results in Figure 26, 27, 28, 29, 30 and 31. Overall, we observe the same phenomena as on Amazon datastes, which further support our claims in the paper.

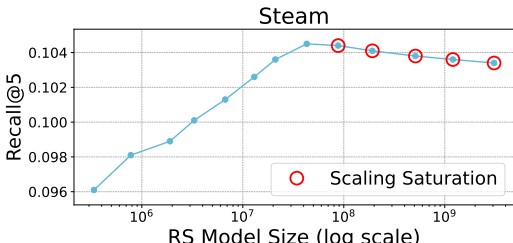

Figure 26: The recommendation performance with varying RS model sizes ($N_{\text{RS}}$) on Steam dataset. The performance quickly saturates as $N_{\text{RS}}$ scales up to $10^7$ parameters.

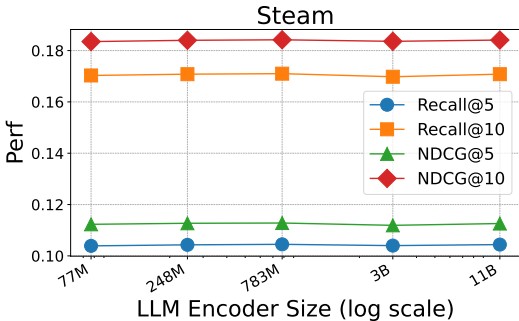

Figure 27: The recommendation performance with varying LLM encoder sizes ($N_{\text{LLM}}$) On Steam dataset. Little to no effective scaling behaviors are observed.

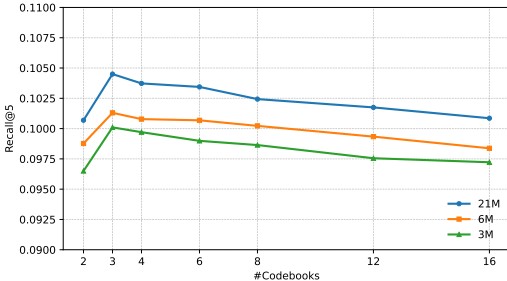

Figure 28: Scaling behaviors of quantization tokenizer when varying the number of codebooks on Steam dataset. Overall, increasing the number of codebooks does not overcome the scaling saturation.

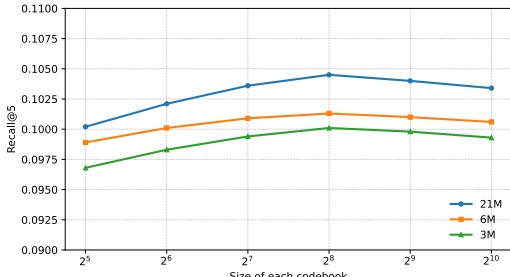

Figure 29: Scaling behaviors of quantization tokenizer when varying the size of each codebooks on Steam dataset. Overall, increasing the number of codebooks does not overcome the scaling saturation.

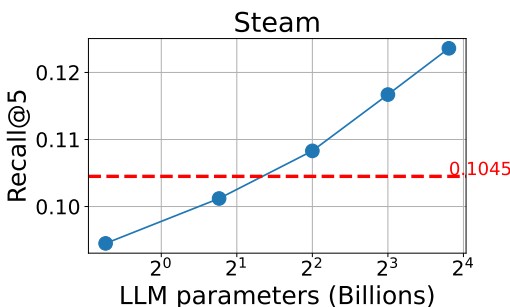

Figure 30: The general scaling behaviors of the LLM-as-RS model ($N_{\text{LoRA}} \approx 0.01 N_{\text{LLM}}$) on Steam dataset. The Red dashed lines are the best scaling performance of the SID-based model.

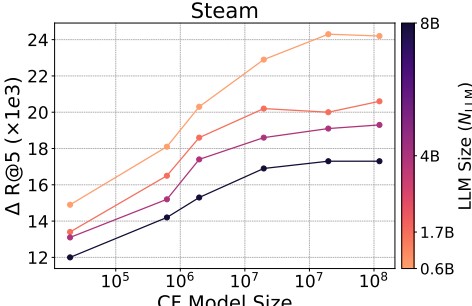

Figure 31: Scaling behaviors of injecting external CF embeddings on Steam dataset. The y-axis metric ($\Delta$Recall@5) measures the performance differences between models with or without external CF embeddings. For a fixed CF model size, the performance gains contributed by its CF embeddings decrease as the LLM backbone size increases, indicating that LLM-as-RS exhibits scaling behaviors for CF signals.

