# OpenReview forum: "Understanding Generative Recommendation with Semantic IDs from a Model-scaling View"
_ICLR.cc/2026/Conference — Submitted to ICLR 2026_

### Official Review · Reviewer_mLTN · 2025-10-28

**Soundness:** 2
**Presentation:** 3
**Contribution:** 3
**Rating:** 6
**Confidence:** 4

**Summary:**

This paper compares two AI recommendation methods: SID-based GR and LLM-as-RS. SID-based GR hits a performance ceiling quickly when scaling up. LLM-as-RS keeps improving with larger models, achieving 20% better results. The authors also show that LLMs can effectively learn user behavior patterns and improve with scale, making LLM-as-RS more promising for future recommendation systems despite higher computational costs.

**Strengths:**

1. The paper addresses one of the most critical questions in the recommender systems field: which paradigm should generative recommendation follow? The research question demonstrates both novelty and necessity, tackling a fundamental challenge in the current landscape of recommendation systems.

2. The authors conduct extensive experiments and derive numerous valuable insights that will benefit future research in this domain. The empirical findings provide meaningful guidance for the community.

3. The paper is well-written and easy to follow, making complex concepts accessible to a broad audience.

**Weaknesses:**

1. The model employs an encoder-decoder architecture (TIGER) for the SID approach. However, encoder-decoder architectures inherently suffer from significant scaling limitations compared to decoder-only LLMs. The authors do not acknowledge this fundamental constraint, which could confound their conclusions about SID's scaling behavior. Note: this refers to the recommender architecture, not the text embedding component.
2. All experiments are conducted exclusively on Amazon datasets. To strengthen the generalizability of the findings, the study should include at least one dataset from a non-Amazon platform. This would provide more convincing evidence for the claimed conclusions.
3. The datasets used are relatively small-scale academic benchmarks, yet the authors make broad claims about the entire community's research direction. Given the significance of these claims, industrial-scale datasets and ideally online A/B test results should support such conclusions. The lack of large-scale validation limits the impact of the findings.
4. The paper discusses performance gaps between SID and LLM-as-RS approaches but lacks a fair comparison at equivalent parameter scales. LLM-as-RS models range from billions to 14 billion parameters, while the largest SID model is only 192M parameters. For an empirical study focused on scaling behavior, comparing models at similar scales is essential. The authors should provide controlled experiments with parameter-matched models.

**Questions:**

1. The paper concludes that SID remains meaningful primarily due to efficiency advantages. Have the authors conducted any experiments to provide insights on how to improve SID's performance? What strategies should be explored to enhance SID models' scaling capabilities from a scaling perspective?
2. The efficiency section shows that LLM4Rec's inference speed is substantially slower than SID, suggesting it may be impractical for real-world industrial applications. Given that LLM4Rec is positioned as the primary recommended architecture, how do the authors propose addressing these practical deployment challenges?
3. The paper does not explain how embeddings are extracted in the SID approach. Could the authors clarify this process?
4. Have the authors tested their findings on other datasets? Do the results generalize beyond the Amazon datasets, or are there domain-specific patterns that affect the conclusions?

---

> ### Author Response · Authors · 2025-11-22
>
> >**W1** The model employs an encoder-decoder architecture (TIGER) for the SID approach. However, encoder-decoder architectures inherently suffer from significant scaling limitations compared to decoder-only LLMs. The authors do not acknowledge this fundamental constraint, which could confound their conclusions about SID's scaling behavior. Note: this refers to the recommender architecture, not the text embedding component.
>
>
>
> **Answer.** Thank you very much for your comment! We have tested the scaling behaviors of decoder-only recommendation model. And in general it has worse model scaling behaviors than the encoder-decoder models. We list their model scaling performance with metric Recall@5 on Toys dataset for comparsion in the table below. Moreover, the conclusion is also confirmed by open-source tooling and experiments in [1].
>
> |  Model Size | 3.3M  | 6.7M  | 13M  | 21M  | 43M  | 88M  |
> |-----|-----|-----|-----|-----|-----|-----|
> | Decoder-only| 0.0352 | 0.0366 | 0.0377 | 0.0375 | 0.0373 | 0.0370 |
> | Encoder-Decoder| 0.0386| 0.040| 0.0409| 0.0409| 0.0407| 0.0402 |
>
> [1] Generative Recommendation with Semantic IDs: A Practitioner’s Handbook
>
>
> >**W2** All experiments are conducted exclusively on Amazon datasets. To strengthen the generalizability of the findings, the study should include at least one dataset from a non-Amazon platform. This would provide more convincing evidence for the claimed conclusions.
>
> **Answer.** Thank you very much for your comment! To provide more comprehensive results, we repeat the results in Steam dataset[1], which does not belong to the Amazon datasets. **We updated the complete results in the manuscript in Appendix O**. Overall, we observe the same phenomena as on Amazon datastes, which further support our claims in the paper.
>
> [1] Self-attentive sequential recommendation
>
> >**W3** The datasets used are relatively small-scale academic benchmarks, yet the authors make broad claims about the entire community's research direction. Given the significance of these claims, industrial-scale datasets and ideally online A/B test results should support such conclusions. The lack of large-scale validation limits the impact of the findings.
>
> **Answer.** Thank you very much for your comments! We would like to first clarify that we do not position that the entire community should shift research directions. In fact, we believe SID-based generative recommendation is also a useful and parameter-efficient path to pursue scaling. However, as a model family, it may eventually be eclipsed by the LLM-as-RS paradigm. This point is discussed in Section 5 and illustrated in Figure 7: the crossover likely depends on multiple factors such as data scale, task complexity, and other practical considerations. But at current stage, SID-based GR models remain the more efficient—and often preferred—choice.
>
> Moreover, one goal of the scaling study is to investigate the model's properties on small scales and extraplote the conclusions to build large models, since the experiments costs of directly traning on large scales are usually unaffordable [1,2,3,4]. Due to the budget limit and risk-control consideration, directly deploying a billion-scale model in industrial senario is not feasible. However, we have conducted rigorous on inverstigated datasets. and we believe the consistency of these results indicates that our conclusions can generalize to larger datasets.
>
> [1] Training Compute-Optimal Large Language Models
> [2] Scaling Laws for Neural Language Models
> [3] Scaling Data-Constrained Language Models
> [4] Scaling Laws with Vocabulary: Larger Models Deserve Larger Vocabularies
>
> >**W4** The paper discusses performance gaps between SID and LLM-as-RS approaches but lacks a fair comparison at equivalent parameter scales. LLM-as-RS models range from billions to 14 billion parameters, while the largest SID model is only 192M parameters. For an empirical study focused on scaling behavior, comparing models at similar scales is essential. The authors should provide controlled experiments with parameter-matched models.
>
> **Answer.** Thank you very much for your comments! To provide a more comprehensive analysis, we also increase the trainable parameter size of SID-based models to 3B. The corresponding results are reported in the table below. We observe that further enlarging the recommendation module does not yield additional performance gains, which strengthens our conclusions.
>
> |  Model Size | 512M  | 1.2B  | 3B  |
> |-----|-----|-----|-----|
> | Beauty |0.0439| 0.0440| 0.0437 |
> | Sports |  0.0244| 0.0243| 0.0241 |
> | Toys | 0.0401| 0.0398| 0.0397 |

---

> ### Author Response · Authors · 2025-11-22
>
> > **Q1** The paper concludes that SID remains meaningful primarily due to efficiency advantages. Have the authors conducted any experiments to provide insights on how to improve SID's performance? What strategies should be explored to enhance SID models' scaling capabilities from a scaling perspective?
>
> **Answer.** Thank you very much for your insightful question! One way to leverage semantic IDs is through alignment, and several very recent works [1,2] have explored this technical direction. The general idea is to first train an LLM to reproduce item content from its semantic IDs, and then train the LLM to predict semantic IDs from a user history represented as a sequence of semantic IDs. This design shortens the decoding process, which can improve inference speed.
>
> We also conduct experiments following this approach on the Beauty dataset using Qwen models, where the LoRA parameters are set to 1% of the total LLM weights. The results (with metric Reacll@5), presented in the table below, show clear scaling behavior but still do not surpass the performance of pure textual LLM-as-RS. This suggests that the approach is promising, though further exploration is needed.
>
> | Category | Qwen3-0.6B | Qwen3-1.7B | Qwen3-4B | Qwen3-8B |
> |----------|------------|------------|----------|----------|
> | Beauty   | 0.0379     | 0.0401     | 0.0416   | 0.0430   |
>
>
> [1] PLUM: Adapting Pre-trained Language Models for Industrial-scale Generative Recommendations, Google, 2025.10
> [2] OneRec-Think: In-Text Reasoning for Generative Recommendation, Kuaishou, 2025.10
>
>
>
> > **Q2** The efficiency section shows that LLM4Rec's inference speed is substantially slower than SID, suggesting it may be impractical for real-world industrial applications. Given that LLM4Rec is positioned as the primary recommended architecture, how do the authors propose addressing these practical deployment challenges?
>
> **Answer:** Thank you for your good question! How to speed up LLM RS is also one of the topics we currently focus on. Based on our understanding, we think the key design should be **asymmetric decoding**. This means that we can use the whole item's contents information as input to let the LLM understand item semantics and keep the scaling properties. But during decoding, the LLM only needs to output several keywords to match the target item. In this way, the model can be both effective and efficient. Moreover, there are many decoding acceleration methods that can be deployed in LLM-as-RS system to improve efficiency, such as speculative decoding [1], layer skipping, and parallel decoding [3].
>
> [1] Efficiency Unleashed: Inference Acceleration for LLM-based Recommender Systems with Speculative Decoding
> [2] LayerSkip: Enabling Early Exit Inference and Self-Speculative Decoding
> [3] FocusLLM: Scaling LLM’s Context by Parallel Decoding
>
>
> >**Q3** The paper does not explain how embeddings are extracted in the SID approach. Could the authors clarify this process?
>
> **Answer.** Thank you very much for the question! We are pleasant to introduce the process: there is a descriptions for each item in the raw dataset, including information like title, brand, price and so on. We directly input the whole description for each item into an embedding LLM such as sentence-t5, the LLM will generate an embedding. In this way, we can generate an embedding for each item.
>
>
> **Q4** Have the authors tested their findings on other datasets? Do the results generalize beyond the Amazon datasets, or are there domain-specific patterns that affect the conclusions?
>
> **Answer.** Please refer to our response to Weakness 2(W2).

---

> > ### Comment · Reviewer_mLTN · 2025-11-28
> >
> > I appreciate the authors' efforts in addressing my concerns. Based on the rebuttal, I will be increasing my score. I strongly suggest including the new experimental results in the final revision. Best of luck with your submission

---

### Official Review · Reviewer_adPM · 2025-10-30

**Soundness:** 2
**Presentation:** 3
**Contribution:** 2
**Rating:** 2
**Confidence:** 4

**Summary:**

This paper foucuses on the performance of semantic id-based generative recommendation (SID-based GR) and LLMs as recommeders (LLM-as-RS) when LLM backbone is scaling up. The authors conducted extensive experiments on multiple datasets with various experiment settings and further lead to the conclusion that compared to LLM-as-RS, SID-based GR shows earlier performance saturation.

Contributions:

**1.** This paper conducted extensive experiments to demonstarte that SID-based GR paradigm shows intrinsic bottleneck when scaling up its components.

**2.** This paper discusses two generative recommendation paradigms from the persprective of the ability of LLMs to capture CF and semantic information.

**Strengths:**

1. The authors clearly define the question discussed and show the experiment results in an organized manner. Also, this paper contains sufficient results figures with elaborate descriptions, showing great writing clarity.

2. This research discusses the two trending generative recommendation paradigms from a novel perspective of model-scaling analysis and further obtain insightful conclusions.

**Weaknesses:**

1. The paper uses models from 336K to 192M parameters conduct scaling experiments of LLM backbone in the SID-based GR. However, one can easily find much larger models, and such small model can hardly deny the scaling law in SID-based GR.

2. When conducting scaling experiments of LLM backbone in the LLM-as-RS, the authors use models from 0.6B to 14B parameters, which is much larger than the model used in experiments of SID-based GR. Existing methods have already applied LLMs of this scale in SID-based GR. There is no reason not using these LLMs.

3. When conducting experiments of LLM encoder and codebook size in SID-based GR, the authors summarized great observations from the results. However, one would more expect insightful analysis from the experiments, not just observations.

**Questions:**

Please refer to the weaknesses.

---

> ### Author Response · Authors · 2025-11-22
>
> >**W1 & W2** The paper uses models from 336K to 192M parameters conduct scaling experiments of **LLM backbone** in the SID-based GR. However, one can easily find much larger models, and such small model can hardly deny the scaling law in SID-based GR.
> When conducting scaling experiments of **LLM backbone** in the LLM-as-RS, the authors use models from 0.6B to 14B parameters, which is much larger than the model used in experiments of SID-based GR. Existing methods have already applied LLMs of this scale in SID-based GR. There is no reason not using these LLMs.
>
> **Answer.** Thank you for your comments. We would like point out that **we have the experiments of LLMs of different sizes in the main text**. They are presented in **Section 3.2 and Figure 3**. Furthermore, we provide more experiments on different LLM encoders in SID-based models in **Appendix G** in the original submission. We also explore using LLM as RS model with SIDs inputs in **Appendix L**.
>
> As for the models from 336K to 192M parameters, they are actually the recommendation model (RS module in Figure 1) for SID-based GR trained from scratch, not the LLM backbone. We also scale the RS module up to 3B, and the results with metric Recall@5 below still support our conclusions: **neither LLM encoder (used to embed items to produce SIDs) nor the RS module (autoregressive model to decode SIDs) shows promising model scaling properties**.
>
> |  Model Size | 512M  | 1.2B  | 3B  |
> |-----|-----|-----|-----|
> | Beauty |0.0439| 0.0440| 0.0437 |
> | Sports |  0.0244| 0.0243| 0.0241 |
> | Toys | 0.0401| 0.0398| 0.0397 |
>
>
>
> > **W3** When conducting experiments of LLM encoder and codebook size in SID-based GR, the authors summarized great observations from the results. However, one would more expect insightful analysis from the experiments, not just observations.
>
> **Answer.** We thank the reviewer for this comment and the opportunity to clarify. We respectfully disagree that the LLM–encoder and codebook–size experiments only provide “observations”; they are explicitly designed to diagnose *where* model scaling fails in SID-based GR.
>
> In Sec. 3.2, we tie the LLM-encoder experiments directly to the scaling law in Eq. (3): by scaling $N_{\text{LLM}}$ from 77M to 11B and showing that performance plateaus, we conclude $\gamma_1 \approx 0$, i.e., **encoder capacity does not translate into additional effective semantic information within SIDs**. Or in plain words, we cannot squeeze more performance out of stronger LLM encoder models within the SID-based framework, prohibiting scaling. Sec. 3.3 then varies both the number and size of codebooks while jointly scaling the RS module; the resulting “sweet spot” at 3×256 and degradation beyond that lead to **the insight that more expressive tokenizers actually *hurt* by producing longer, harder-to-learn SID sequences**, again yielding $\gamma_2 \approx 0$.
>
> Finally, Sec. 3.4 uses an ablation that bypasses SIDs and feeds dense LLM embeddings directly into the RS to test whether the bottleneck is architectural rather than parametric; the gains from this setup **support our claim that the SID interface itself discards crucial semantic information and fundamentally limits scaling**. We will make these causal interpretations more explicit in the revision.

---

> > ### Comment · Reviewer_adPM · 2025-11-24
> >
> > Dear authors, first of all, thanks for your reply to my questions and additional results.
> >
> > Respectfully I want to clarify that in my comment, the term "LLM backbone" refers to the RS module of SID-based GR. I'm willing to raise my score to 4, while my main concern, i.e., the LLM backbone used for SID-based GR being too small to prove your main conclusions, is answered by additional results. However, encoder-decoder architectures suffer from intrinsic problem of scaling, and current works of SID-based GR shows scaling ability with decoder-only architecture [1], which contrasts to your major conclusions. Hence, I think 4 is a reasonable rating for the current version of this work.
> >
> > [1]MiniOneRec: An Open-Source Framework for  Scaling Generative Recommendation X Kong, 2025.

---

> ### Author Response · Authors · 2025-11-24
> **Discussions on MiniOneRec: Why it does not Conflict with Our Findings**
>
> Thank you very much for your clarification and timely reply!
>
> We would first like to respectfully note that MiniOneRec was only posted on arXiv in October, which is **later than ICLR full-paper deadline**. According to ICLR policy, we were therefore not expected to be aware of it or to discuss it in our submission. However, we have recently examined MiniOneRec under the same experimental setting as our paper and also conducted relevant experiments in the rebuttal. We are more than pleased to give **an in-depth discussion here to show that the results in MiniOneRec paper actually does not contradict with our conclusions**.
>
> ### **Central Clarifications:**
> **(1) MiniOneRec only reports the scaling behaviors of LLM RS, which aligns with our observations. Moreover, we have reported scaling behaviors of SID+LLM RS method in Appendix L of the original submission, which is earlier than MiniOneRec paper. (Part 2 below)**
>
> **(2) It introduces a new SID-text alignment task to mitigate the intrinsic bottleneck of SIDs, which proves our claim from a different perspective. (Part 3 & 5 below)**
>
>
> ## **1. Design and Setting of MiniOneRec**
>
> First, we give a high-level summary of MiniOneRec, which is essential for discussions below. MiniOneRec is a **SID+LLM RS** method. Hence, it still contains **two major training stages: SID generation and LLM finetuning**. An important design is that in addition to the usual **next-item prediction** task, MiniOneRec adds a new task in the LLM finetuning stage: **SID-text alignment** that requires LLM to reconstruct the item textual descriptions, which we will further discuss. We list the detailed model comparisons in Table 1. For the following parts, we conduct detailed ablation studies to provide understanding of MiniOneRec's scaling behaviors. All reported results use Recall@5 as metric.
>
> **Table 1. Comparisons between MiniOneRec and models in the paper.**
> ||Item Representations|RS Backbone|Training/Finetuning Tasks | #Training Stage |
> |---|---|---|---|---|
> |SID-based Model in the paper|SIDs|Train-from-scratch Transformer|Next item prediction|Two-Stage|
> |LLM RS| Texts|LLM|Next item prediction|One-Stage|
> |MiniOneRec|SIDs|LLM|Next item prediction & SID-text alignment| Two-Stage at least|
>
> ## **2. Scaling Behaviors in MiniOneRec Paper**
>
> In the MiniOneRec paper, the scaling behaviors only concern **scaling LLM RS in the second stage (RS training)**. In fact, **we have reported similar scaling behaviors of LLM RS when using SIDs as input in Appendix L of the original submission, even earlier than MiniOneRec**. We include the results in Table 2 below for your convenience. In the main text, we have also shown the RS backbone always show scaling behaviors, for both train-from-scratch transformer (Section 3.1) and pretrained LLM (Section 4.1). Hence, the results in MiniOneRec paper actually align with our paper.
>
> **Table 2. The scaling performance of LLM RS with SID inputs (from Appendix L of original submission)**
> | Category | Qwen3-0.6B | Qwen3-1.7B | Qwen3-4B | Qwen3-8B |
> |----------|------------|------------|----------|----------|
> | Beauty   | 0.0356     | 0.0379     | 0.0393   | 0.0409   |
> | Sports   | 0.0197     | 0.0217     | 0.0225   | 0.0234   |
> | Toys     | 0.0367     | 0.0380     | 0.0392   | 0.0398   |
>
>
> ## **3. Our claim about the SID-based Model**
>
> Furthermore, we would like to clarify that our claim on SID-based models **is not** that they do not have scaling behaviors. **Our main claim and key insight is that the SID mechanism has limitations in effectively encoding item-semantic information for recommendation, which can harm the model’s performance and scaling properties.** One key evidence is that when enlarging LLM encoder (to provide item embeddings of more semantic information), the quality of SIDs will not be improved and there are no gains for the downstream tasks (Section 3.2). Our claim focuses on **the first stage (SID generation)**, which does not contradict with MiniOneRec which only reports the scaling behaviors of the second stage.
>
> Moreover, we report the results of using LLM (Qwen3-8B, even larger than the model in MiniOneRec) as RS and scaling the LLM encoder (Flan-T5 series, which has the best performance in our experiments) for SID generation. From the results in Table 3 below, we can see that scaling up LLM encoder still brings no performance gains, **no matter the RS backbone is LLM or train-from-scratch transformer**. This further indicates that the SID mechanism acts as a bottleneck, reinforcing our central claim.
>
> **Table 3. Scaling LLM encoder for SID generation when using Qwen3-8B as RS backbone. No performance gain is observed.**
> | Model Size| 77M | 248M | 783M | 3B | 11B|
> |---------- |------------|------------|----------|----------|---|
> | Beauty    | 0.0405     | 0.0409     | 0.0407   | 0.0409   |0.0408|
> | Sports    | 0.0229     | 0.0232     | 0.0235   | 0.0234   |0.0234|
> | Toys      | 0.0395     | 0.0400     | 0.0397   | 0.0398   |0.0399|

---

> ### Author Response · Authors · 2025-11-24
> **Discussions on MiniOneRec: Why it does not Conflict with Our Findings (Continue)**
>
> ## **4. Does Decoder-only Model Necessarily Have Better Scaling Properties than Encoder-Decoder Model?**
>
> Reviewer-mLTN also noted this point, and we include the relevant results here for reference. We compare the scaling behaviors of training from scratch with both encoder–decoder and decoder-only architectures. Overall, the decoder-only models exhibit weaker scaling behaviors than the encoder–decoder models. For comparison, we summarize their Recall@5 performance on the Toys dataset in Table 4. Moreover, this conclusion is further supported by open-source tooling and experiments in [1].
>
> Therefore, we conclude that **decoder-only models do not necessarily exhibit better scaling behaviors than encoder–decoder models on recommendation tasks**. The **superior performance of LLM-based recommendation systems should be attributed to their generalizability, rather than merely their encoder–decoder structure**.
>
>
> **Table 4. Ablation Study on Transformer Arch**
> |  Model Size | 3.3M  | 6.7M  | 13M  | 21M  | 43M  | 88M  |
> |-----|-----|-----|-----|-----|-----|-----|
> | Decoder-only| 0.0352 | 0.0366 | 0.0377 | 0.0375 | 0.0373 | 0.0370 |
> | Encoder-Decoder| 0.0386| 0.040| 0.0409| 0.0409| 0.0407| 0.0402 |
>
> [1] Generative Recommendation with Semantic IDs: A Practitioner’s Handbook
>
>
> ## **5. Source of MiniOneRec's Performance.**
>
> Last but not least, we show that the strong performance of MiniOneRec is largely attributable to the SID–text alignment (STA) task. We conducted an ablation study on the Beauty dataset using three configurations: **“SID + LLM RS without STA,” “SID + LLM RS with STA,”** and **“Pure Texts + LLM RS.”** The results are presented in Table 5.
>
> We observe that SID–text alignment brings substantial performance improvements to the “SID + LLM RS” setup. **This provides evidence for our claim that the SID mechanism has inherent limitations in effectively encoding item-semantic information from another perspective**. Because if the original SIDs already carried sufficiently rich semantic information, then the STA task would be redundant and would not yield such significant gains.
>
> Even so, the STA task can only partially compensate for the information loss introduced during SID generation. **Its performance still lags behind that of “Pure Texts + LLM RS,” which further strengthens our argument.**
>
> We hope this thorough discussion helps clarify the design of MiniOneRec and shows that it is not in conflict with our findings. In fact, we reported similar results and insights even earlier than the MiniOneRec paper.
>
> **Table 5. Ablation Study on SID-text Alignment Task**
> | Category | Qwen3-0.6B | Qwen3-1.7B | Qwen3-4B | Qwen3-8B |
> |----------|------------|------------|----------|----------|
> |SID+LLM RS without STA| 0.0356     | 0.0379     | 0.0393   | 0.0409   |
> | SID+LLM RS with STA  | 0.0379     | 0.0401     | 0.0416   | 0.0430   |
> | Pure texts+LLM RS     | 0.0385     | 0.0422     | 0.0449   | 0.0475   |

---

> > ### Comment · Reviewer_adPM · 2025-11-27
> >
> > Thanks for thorough analysis and additional results. However, I will keep my score of 4. First, in part 4 of your reply, you utilized decoder-only architectures with model size varying from 3.3M to 88M, which is rather small model size to prove model scaling ability as mentioned in the original comments. Second, as mentioned in Appendix L of the manuscript, putting SIDs with task description as input sequence of LLM shows scaling ability. However, this paradigm is generally viewed as SID-based GR in current works[1, 2]. From this point, your main conclusion seems less insightful. Hence, I think 4 is a reasonable rating. Best wishes and good luck.
> >
> > [1] Adapting Large Language Models by Integrating  Collaborative Semantics for Recommendation. ICDE 2024.
> >
> > [2] Unleash LLMs Potential for Sequential Recommendation by Coordinating Dual Dynamic Index Mechanism. WWW 2025.

---

### Official Review · Reviewer_gVU9 · 2025-10-31

**Soundness:** 2
**Presentation:** 3
**Contribution:** 2
**Rating:** 4
**Confidence:** 4

**Summary:**

This paper presents a empirical study on the scaling behaviors of modern generative recommendation (GR) paradigms. The authors focus on two representative frameworks:
- SID-based GR, where items are represented as semantic IDs (SIDs) derived from modality encoders and quantization tokenizers.
- LLM-as-RS, where large language models (LLMs) are directly used to generate recommendations without intermediate SIDs.
The paper found that SID-based GR exhibits early performance saturation while LLM-as-RS continues to improve with model size. The authors argue that LLM-as-RS represents a more scalable and semantically expressive path toward developing foundation models for recommendation systems.

**Strengths:**

**S1:Scope**: The paper focuses on an important topic of generative recommendation systems: scaling behaviors. Understanding how different GR paradigms scale with model size is crucial for building effective recommendation foundation models.

**S2:Experiment**"" Empirical study across model sizes and LoRA configurations provides valuable evidence for analyzing LLM-as-RS scaling trends.

**Weaknesses:**

**W1: Limited analysis on data scaling**: While the paper provides a thorough analysis of model scaling, it lacks a comprehensive examination of data scaling [1]. The training data used in the experiments is small-scale (1M interactions). As far as I know, SASRec with 8e6 parameter are saturated on these datasets. And as shown in Figure 2, Generative recommendation models performance saturate after parameter size reaches 1e7. Without dataset scaling analysis, it is hard to ablate whether the performance saturation is due to model size or data size.

**W2: Semantic and CF information are not clearly disentangled in the scaling analyse on LLM-as-RS**: "Throughout these experiments, the trainable LoRA weights are maintained at approximately 1% of the total LLM parameters." As a result, when scaling up the backbone LLM, both the representational capacity (semantic understanding of items) and the effective amount of trainable parameters grow simultaneously, making it difficult to attribute the observed performance improvements to either factor. To truly disentangle these effects, it would be necessary to compare models of different backbone sizes under varying LoRA capacities (e.g., fixed LoRA rank vs. proportional LoRA rank), and analyze whether scaling gains primarily stem from the frozen LLM’s semantic prior or from the additional trainable parameters. Without such a control, the interpretation of γ and β as independent scaling coefficients for semantic and CF learning remains ambiguous.

**W3: LLM-as-RS’s CF scaling claim appears internally inconsistent.**: The paper presents seemingly conflicting evidence regarding the scaling of LLMs’ ability to capture collaborative filtering (CF) signals.
In Section 3, the authors argue that SID-based GR models fail to scale effectively, largely because their CF-driven representations saturate quickly.
However, in Section 4.2, they employ larger SASRec models to provide “stronger” CF embeddings and observe that larger LLM-as-RS models benefit less from these external CF signals.
If CF signals themselves are bounded and non-scalable—as implied by the earlier results—it remains unclear how the authors justify that the LLM-as-RS’s improved scaling stems from its enhanced ability to model CF information.

[1] Hoffmann, Jordan, et al. "Training compute-optimal large language models." arXiv preprint arXiv:2203.15556 (2022).

**Questions:**

Refer to Weaknesses.

---

> ### Author Response · Authors · 2025-11-22
>
> >**W1**: Limited analysis on data scaling: While the paper provides a thorough analysis of model scaling, it lacks a comprehensive examination of data scaling [1]. The training data used in the experiments is small-scale (1M interactions). As far as I know, SASRec with 8e6 parameter are saturated on these datasets. And as shown in Figure 2, Generative recommendation models performance saturate after parameter size reaches 1e7. Without dataset scaling analysis, it is hard to ablate whether the performance saturation is due to model size or data size.
>
> **Answer.** Thank you very much for your comment! From the perspective of scaling laws, the saturation points of scaling curves are **determined jointly by the model size and dataset size** [1]. Specifically, for given dataset size D, the model scaling curve will saturate when the model size N is larger than $aD^b$. Hence, one approach to compare the scaling properties of two models is to check which one saturates lower when increasing the parameters. Our experiments aim to show that on the same dataset, the LLM-as-RS can still have performance gains from scaling up when the SID-based model already saturates. This is one evidence that LLM-as-RS is has better scaling behaviors.
> Furthermore, we compare the data scaling behavior of the SID-based model and LLM-as-RS on Beauty dataset. From the table below, we find that LLM-as-RS has better scaling behaviors, which enhances our conclusions.
>
> |  Data Ratio | 20%  | 40%  | 60%  | 80%  | 100%  |
> |-----|-----|-----|-----|-----|-----|
> | SID-based | 0.0395 | 0.0410 | 0.0429 | 0.0442 | 0.0453 |
> | LLM-as-RS | 0.0451 | 0.0472 | 0.0487 | 0.0505 | 0.0515 |
>
> [1] Scaling Laws for Neural Language Models
>
>
> >**W2**: Semantic and CF information are not clearly disentangled in the scaling analyse on LLM-as-RS: "Throughout these experiments, the trainable LoRA weights are maintained at approximately 1% of the total LLM parameters." As a result, when scaling up the backbone LLM, both the representational capacity (semantic understanding of items) and the effective amount of trainable parameters grow simultaneously, making it difficult to attribute the observed performance improvements to either factor. To truly disentangle these effects, it would be necessary to compare models of different backbone sizes under varying LoRA capacities (e.g., fixed LoRA rank vs. proportional LoRA rank), and analyze whether scaling gains primarily stem from the frozen LLM’s semantic prior or from the additional trainable parameters. Without such a control, the interpretation of γ and β as independent scaling coefficients for semantic and CF learning remains ambiguous.
>
>
> **Answer.** Thank you for the comment. We would like to clarify some of our settings and experiments here.
> (1) The quote in the review is **specific to the experiments shown in Figure 7**. For other experiments on LLM-as-RS, we do not fix LoRA weights as one percent of the totally weights. We are sorry if the expression caused confusion. We have fixed them in the manuscript.
>
> (2) For the experiments required for "different backbone sizes under varying LoRA capacities", we we agree with you these are useful, and **these are actually already provided in the original submission in Figure 8**.  They may have been missed in the original review. In these experiments, we vary the backbone sizes and the LoRA sizes repsectively, which are just the required comparisons of "fixed LoRA rank vs. proportional LoRA rank". We analyze the results of Figure 8 through the lens of model scaling law, showing that both frozen LLM backbone and trainable LoRA weights contribute to the performance gains. Furthermore, we disentangle both $\gamma$ and $\beta$ in Section 4.2 by solely injecting the collaborative signals. The experiments prove that LLM backbones have model scaling behaviors on both learning semantic and CF knowledge, i.e., $\gamma>0$ and $\beta>0$.

---

> ### Author Response · Authors · 2025-11-22
>
> > **W3: LLM-as-RS’s CF scaling claim appears internally inconsistent.:** The paper presents seemingly conflicting evidence regarding the scaling of LLMs’ ability to capture collaborative filtering (CF) signals. In Section 3, the authors argue that SID-based GR models fail to scale effectively, largely because their CF-driven representations saturate quickly. However, in Section 4.2, they employ larger SASRec models to provide “stronger” CF embeddings and observe that larger LLM-as-RS models benefit less from these external CF signals. If CF signals themselves are bounded and non-scalable—as implied by the earlier results—it remains unclear how the authors justify that the LLM-as-RS’s improved scaling stems from its enhanced ability to model CF information.
>
> **Answer.** Thank you for your great comments! We would like to clarify some settings to show there is no inconsistency here.
> (1) First, we would like to point out that we do not claim that "CF signals themselves are bounded and non-scalable" in the paper. Instead, what we show in the section 3 is that SID-based model has scaling behaviors when the model size is not too large. But SID-based GR models have limitations in their ability to extract the CF signals when scaling up.
>
> (2) We agree that SASRec will saturate when its size is larger than a certain threshold value. However, **we use the SASRec models whose sizes that fall in the range that shows scaling behaviors**. We list the model's scaling performance below.
>
> |  Model Size | 0.1M  | 0.8M  | 1.5M  | 6.3M  | 25M  | 75M  |
> |-----|-----|-----|-----|-----|-----|---|
> | Recall@5 | 0.0201 | 0.0213 | 0.0224 | 0.0233 | 0.0240 | 0.0244 |
>
> (3) As a subtle point, we also want to point out that the SID-based recommendation model is an **encoder-decoder** model while SASRec is a **decoder-only** model. They are not of the same structure and should not be viewed as equivalent. Hence, the results in Section 3 should not be directly interpreted as conflicts with experiments in Section 4.
>
> We hope the above clarifications could justify our experiment design and show there is no inconsistency in the experiments. We have also updated the manuscript to make it clearer.

---

### Official Review · Reviewer_rLfN · 2025-11-01

**Soundness:** 3
**Presentation:** 2
**Contribution:** 3
**Rating:** 4
**Confidence:** 4

**Summary:**

The paper studies how generative recommendation (GR) scales along two paradigms: (i) SID-based GR (quantizing item semantics into discrete “semantic IDs” and training a sequential recommender on SID sequences) and (ii) LLM-as-RS (fine-tuning an LLM to directly generate the next item title with constrained decoding). It proposes a scaling-law formulation grounded in a risk decomposition where total error (measured as MR@k = 1−Recall@k) separates into semantic-information (SI) error and collaborative-filtering (CF) error. The resulting expression predicts Recall@k as a sum of two *decreasing power-law* terms in the effective SI/CF parameter budgets.   Experiments use three Amazon subsets—Beauty, Sports & Outdoors, Toys & Games—with fixed splits and preprocessing following prior work; sequence-length statistics are reported.

**Strengths:**

- **Unified scaling-law lens for GR.** The paper provides a *single, interpretable* equation for Recall@k that decomposes SI vs. CF contributions, enabling side-by-side comparison of paradigms and modules under a fixed data budget.
- **Clear paradigm contrast.** It operationalizes both **SID-based** and **LLM-as-RS** within a comparable evaluation protocol, exposing where and why SID-based methods saturate and how LLM-as-RS keeps improving with scale.
- **Concrete, reproducible LLM-as-RS design.** The constrained-decoding setup (trie + beam search) and minimal prompt are precisely described, helping readers *replicate* next-item generation on catalog vocabularies.
- **Negative results are informative.** The analysis doesn’t just report wins; it **identifies bottlenecks** in SID pipelines (limited SID semantic capacity), which is practically valuable for future GR system design.
- **Actionable takeaway for practitioners.** If compute is available, **scaling LLM-as-RS** appears a more reliable path to gains than scaling SID encoders/tokenizers/recommenders alone; additionally, the **diminishing returns of external CF embeddings** at larger LLM sizes guide feature-engineering choices.
- **Dataset transparency.** The paper discloses dataset stats and sequence-length distributions, aligning with the scaling-law narrative and enabling fairer comparisons and efficiency analyses.

**Weaknesses:**

### 1) Metric–task **conflation**: treating “title generation = recall hit”

- The paper reports **Recall@k/NDCG@k**, yet **LLM-as-RS** decodes with **constrained beam search + a trie built from the catalog**, i.e., the **candidate space equals the full catalog** and a “hit” is counted by **exact title string match**.
- This **collapses text generation** into **character-level constrained search over a dictionary**, which is closer to **vocabulary-based retrieval/relabeling** than open-ended generation. Results are extremely sensitive to **synonyms/aliases**, **normalization/cleaning rules**, and **string equivalence**. The paper does not report **title→item-ID coverage/ambiguity**, **dedup/alias policies**, or **failure cases** (e.g., homonyms).

------

### 2) Insufficient **identifiability** and **fit reliability** for the scaling-law formula

- Recall@k is modeled as the **difference of two power-law terms** (Eqs. (2)/(3)) with **γ, β** as effective weights; the appendix later claims **γ₁, γ₂ ≈ 0** (encoder/tokenizer scale has no effect) and proceeds with **simplified fits** (Eqs. (9)/(10)).
- **Identifiability is not discussed**: when multiple module sizes co-vary and performance–scale is fit with a **log–power-law**, parameters (**R₀, A, B, a, b, γ, β**) can be **highly collinear**; multiple parameterizations may yield **high R²** yet **different physical interpretations** (e.g., the claim that the LLM contributes no SI). R²>0.9 alone does **not** validate the **semantics** of the parameters.
  - **Recommendations:**
    - Report **sensitivity analyses/bootstrapped CIs** and **parameter-correlation heatmaps**.
    - Use **regularization** or **hierarchical Bayesian** fitting to mitigate over-fitting.
    - Provide **stability across datasets/seeds** for the fitted parameters.
- The evidence for **β>0** (lower error on a **20% held-out** when using Eq. (4) vs. a weakened Eq. (6)) is **fragile**: the held-out is drawn from the **same curve** (e.g., Fig. 8) rather than an independent setting, and it compares only **two approximations’** prediction errors—**noise-sensitive** by design. Stronger statistical tests (e.g., **multi-split CV**, **likelihood-ratio tests**) are needed.

------

### 3) **Baselines and ablations** are incomplete

- Prior work such as **GenRec (ECIR 2024)** (directly generating the next item name) and **COBRA/LIGER (2024–2025)** (hybrid sparse SID + dense text) are **mentioned** but **not reproduced under the same data/protocol**, making it hard to align the paper’s scaling observations with existing systems.

------

### 4) **Evaluation statistics** and **reproducibility** details are insufficient

- Missing **confidence intervals** or **mean±std over multiple runs**; figures show **no error bars**. Scaling claims typically require **robustness across random seeds** and **subsampled training sets**.

------

### 5) **Writing and figure quality**

- **Spelling/grammar:**
  - “**exihibit** scaling behaviors” → *exhibit*
  - “**argumnet** that β>0” → *argument*
  - Figure 14 title “**Beuaty**” → *Beauty*
  - Figure 10 caption “**structuture**” → *structure*
  - “**There statistics** will affect …” → *These statistics*
- **Symbols/terminology:**
  - Typos such as “**traing** stage” near Eq. (3).
  - Definitions of **γ** and **β** are **scattered**.

**Questions:**

No

---

> ### Author Response · Authors · 2025-11-22
>
> >**W1** The paper reports Recall@k/NDCG@k, yet LLM-as-RS decodes with constrained beam search + a trie built from the catalog, i.e., the candidate space equals the full catalog and a “hit” is counted by exact title string match.
> This collapses text generation into character-level constrained search over a dictionary, which is closer to vocabulary-based retrieval/relabeling than open-ended generation. Results are extremely sensitive to synonyms/aliases, normalization/cleaning rules, and string equivalence. The paper does not report title→item-ID coverage/ambiguity, dedup/alias policies, or failure cases (e.g., homonyms).
>
> **Answer**: Thank you very much for your comment! We would first like to clarify that constrained beam search is a general decoding strategy widely used in mainstream generative recommendation systems. Both SID-based methods [1,2,3] and LLM-as-RS approaches [4,5] employ this strategy during inference.
>
>
> Regarding the title duplication issue, we would like to clarify this issue is not material to our results, because it has already been accounted for during the data preprocessing stage.  Specifically, the duplication rate is minimal when we limit item titles to 10 tokens (the statistics are given in the table below). And in order to disregard even this minimal collision, we replace the final token of the identical pairs with a different token from their item descriptions respectively.  Hence, the reported results are not impacted, and we will be sure to add this context in the paper as we agree it is important to ensure fairness in this comparison.
>
>
> ||Beauty|Sports|Toys|
> |---|---|---|---|
> |#Total Items|12,101|18,357|11,924|
> |#Item pairs with same titles|257|310|156|
>
>
> [1]  Recommender systems with generative retrieval (TIGER)
> [2]  Unified generative recommendations with cascaded sparse-dense representations (COBRA)
> [3]  Unifying generative and dense retrieval for sequential
> recommendation (LIGER)
> [4]  IDGenRec: LLM-RecSys Alignment with Textual ID Learning
> [5]  LLaRA: Large Language-Recommendation Assistant
>
>
> > **W2.1** Identifiability is not discussed: when multiple module sizes co-vary and performance–scale is fit with a log–power-law, parameters (R₀, A, B, a, b, γ, β) can be highly collinear; multiple parameterizations may yield high R² yet different physical interpretations (e.g., the claim that the LLM contributes no SI). R²>0.9 alone does not validate the semantics of the parameters.
> Recommendations: - Report sensitivity analyses/bootstrapped CIs and parameter-correlation heatmaps. - Use regularization or hierarchical Bayesian fitting to mitigate over-fitting. - Provide stability across datasets/seeds for the fitted parameters.
>
>
> **Answer:** Thank you very much for your comment! We would like to clarify that the claims in our paper are supported not only by the fitted scaling parameters. For instance, our observation that “the LLM encoder in the SID-based model shows no scaling behavior” (which differs from the reviewer’s phrasing that “the LLM contributes no SI”) is primarily grounded in the direct empirical result that enlarging the LLM encoder yields no performance gains, together with the fact that in SID-based models the LLM encoder only provides semantic item embeddings. In addition, the scaling-law equations we adopt follow prior studies on scaling laws [1,2]; these formulations and their associated parameters have been validated and are widely accepted by the community.
>
> Moreover, to further improve the reliability of the fitted scaling parameters, we followed the reviewer’s suggestion and evaluated their stability across four different random seeds. Concretely, we ran four trials with different seeds and fit the scaling equation independently for each trial, producing four sets of fitted parameters. We then report the mean and standard deviation of these parameters in the table.
>
> [1] Scaling laws for generative mixed-modal language models, ICML 2023
> [2] Scaling laws with vocabulary: Larger models deserve larger vocabularies, NIPS 2024
>
>
> From the table, we observe that the deviations of the fitted parameters are very small, which demonstrates their stability and supports the reliability of our conclusions.
>
> |          | Beauty | Sports | Toys  |
> |----------|:------:|:------:|:-----:|
> | $R_0$    | 4.5e-1 $\pm$ 2e-2 | 3.1e-1 $\pm$ 1e-2   | 1.7e-1 $\pm$ 1e-2|
> | $A$      | 16.8 $\pm$ 1.1   | 24.4 $\pm$ 1.4   | 6.1 $\pm$ 0.3   |
> | $B$      | 1e-2 $\pm$ 8e-4  | 1e-2 $\pm$ 6.5e-4  | 1e-2 $\pm$ 7e-4 |
> | $a$      | 0.61 $\pm$ 0.01   | 0.62 $\pm$ 0.02  | 0.51 $\pm$ 0.01 |
> | $b$      | 2.24 $\pm$ 0.03  | 1.98 $\pm$ 0.02  | 2.02 $\pm$ 0.01 |

---

> ### Author Response · Authors · 2025-11-22
>
> >**W2.2** The evidence for β>0 (lower error on a 20% held-out when using Eq. (4) vs. a weakened Eq. (6)) is fragile: the held-out is drawn from the same curve (e.g., Fig. 8) rather than an independent setting, and it compares only two approximations’ prediction errors—noise-sensitive by design. Stronger statistical tests (e.g., multi-split CV, likelihood-ratio tests) are needed.
>
> **Answer:** Thank you very much for your comment. To further validate our results, we employ multi-split cross-validation. Specifically, we use 5-fold cross-validation and repeat the entire procedure 5 times. The final averages and standard deviations of the held-out errors are reported in the table. From the table, we observe that the held-out errors for $\beta > 0$ are consistently lower, which further strengthens our conclusion.
>
>
> |        | $\beta = 0$ | $\beta > 0$ |
> |--------|:-----------:|:-----------:|
> | Beauty | 4.1e-4 $\pm$ 2e-5      | **3.2e-4** $\pm$ 1e-5 |
> | Sports | 1.1e-3  $\pm$ 3e-5      | **3.4e-4** $\pm$ 2e-5 |
> | Toys   | 1.2e-3  $\pm$ 4e-5    | **9.1e-4** $\pm$ 2e-5 |
>
>
> > **W3** Prior work such as GenRec (ECIR 2024) (directly generating the next item name) and COBRA/LIGER (2024–2025) (hybrid sparse SID + dense text) are mentioned but not reproduced under the same data/protocol, making it hard to align the paper’s scaling observations with existing systems.
>
>
> **Answer:** Thank you very much for your comment! The primary goal of this work is to understand the fundamental paradigms of generative recommendation through the lens of model scaling. Following prior studies on scaling laws[1,2,3], our focus is on analyzing the basic generative recommendation model, rather than proposing a new state-of-the-art method.
>
> TIGER is the most foundational model among SID-based approaches and serves as the backbone for subsequent methods such as COBRA and LIGER. This is why we selected it as our main subject of study, and we believe that our conclusions are likely to generalize to other SID-based models.
>
> To empirically support this claim, we repeat the scaling experiments on LIGER (as the official implementations of ECIR and COBRA are not available) and present the results in the table below. The observations for LIGER align closely with those from TIGER, further demonstrating the generality and robustness of our conclusions.
>
> **Table 1. RS Model scaling of LIGER: the model still saturates fast**
>
> |  Model Size | 3.3M  | 6.7M  | 13M  | 21M  | 43M  | 88M  | 192M  |
> |-----|-----|-----|-----|-----|-----|-----|-----|
> | Recall@5 | 0.0441 | 0.0456 | 0.0467 | 0.0475 | 0.0473 | 0.0470 | 0.0469 |
>
>
>
> **Table 2. LLM encoder scaling of LIGER: No scaling behaviors**
>
> |  LLM Size | 77M  | 248M  | 783M  | 3B  | 11B  |
> |-----|-----|-----|-----|-----|-----|
> | Recall@5 | 0.0470 | 0.0472 | 0.0471 | 0.0475 | 0.0474 |
>
> **Table 3. Scaling the number of codebooks: No scaling behaviors**
>
> |  #Codebooks | 2  | 3  | 4  | 6  | 8  |
> |-----|-----|-----|-----|-----|-----|
> | Recall@5 | 0.0467 | 0.0475 | 0.0471 | 0.0463 | 0.0460 |
>
> **Table 4. Scaling the size of each codebooks: No scaling behaviors**
>
> |  Size of each codebook | 32  | 64   | 256  | 512  | 1024  |
> |-----|-----|-----|-----|-----|-----|
> | Recall@5 | 0.0469 | 0.0473 | 0.0475  | 0.0475 | 0.0472 |
>
> [1] When Scaling Meets LLM Finetuning: The Effect of Data, Model, and Finetuning Method
> [2] Scaling Laws for Generative Mixed-Modal Language Models
> [3] Scaling Data-Constrained Language Models
>
> > **W4** Missing confidence intervals or mean±std over multiple runs; figures show no error bars. Scaling claims typically require robustness across random seeds and subsampled training sets.
>
> **Answer:** Thank you very much for your comment! We ran 3 runs per experiment with different seeds, and observed **minimal variance with average of 3e-4**. We have updated the error bars in the main text. For the error bars in the appendix, due to resource limitation, we will update them in a future version.
>
>
> > **W5.** Writing and figure quality
>
> **Answer:** Thank you very much for pointing out the typos! We have fixed them and updated the manuscript.

---

### Meta-Review · Area_Chair_oTta · 2026-01-03

**Summary:**

The reviewers had concerns regarding the validity of the model architecture used for empirical analysis, the completeness of the experiments, and the lack of insightful analysis of the empirical observations.

The authors' rebuttals clarified some issues and provided additional experimental results, but unfortunately, they did not address the reviewers' key concerns regarding the previously mentioned issues. So, I think it's a weak reject.

**Reviewer Concerns:**

Reviewer rLfN and mLTN's concerns are primarily addressed.

Reviewer gVU9 and adPM's concerns regarding the validity of the model architecture used for empirical analysis, completeness of the experiments, and lack of insightful analysis of the empirical observations remain outstanding.

**Reviewer Scores:**

Reviewer rLfN might raise his score.
Reviewer gVU9 and adPM's concerns are critical, even if they changed the score, which will still be weak reject.
Reviewer mLTN promised to rasie his score to 8.

---

### Decision · Program_Chairs · 2026-01-26

Reject